# Family as a health promotion setting: A scoping review of conceptual models of the health-promoting family

**Valerie Michaelson**[1]*, **Kelly A. Pilato**[1], **Colleen M. Davison**[2]

**1** Department of Health Sciences, Brock University, St. Catharines, Canada, **2** Department of Public Health Sciences, Queen's University, Kingston, Canada

* vmichaelson@brocku.ca

**Data Availability Statement:** All relevant data are within the paper and its Supporting Information files.

**Funding:** Support for this analysis included one operating grant (CIHR Grant MOP 341188,

## Abstract

### Background

The family is a key setting for health promotion. Contemporary health promoting family models can establish scaffolds for shaping health behaviors and can be useful tools for education and health promotion.

### Objectives

The objective of this scoping review is to provide details as to how conceptual and theoretical models of the health promoting potential of the family are being used in health promotion contexts.

### Design

Guided by PRISMA ScR guidelines, we used a three-step search strategy to find relevant papers. This included key-word searching electronic databases (Medline, PSycINFO, Embase, and CINAHL), searching the reference lists of included studies, and intentionally searching for grey literature (in textbooks, dissertations, thesis manuscripts and reports.)

### Results

After applying inclusion and exclusion criteria, the overall search generated 113 included manuscripts/chapters with 118 unique models. Through our analysis of these models, three main themes were apparent: 1) ecological factors are central components to most models or conceptual frameworks; 2) models were attentive to cultural and other diversities, allowing room for a wide range of differences across family types, and for different and ever-expanding social norms and roles; and 3) the role of the child as a passive recipient of their health journey rather than as an active agent in promoting their own family health was highlighted as an important gap in many of the identified models.

received by VM and CM) and one project grant
(CIHR Grant MOP 97962, received by CM and VM),
both from the Canadian Institutes of Health
Research. The funders did not play any role in the
study design, data collection and analysis, decision
to publish, or preparation of the manuscript.
https://cihr-irsc.gc.ca/e/193.html.

**Competing interests:** The authors have declared
that no competing interests exist.

## Conclusions

This review contributes a synthesis of contemporary literature in this area and supports the priority of ecological frameworks and diversity of family contexts. It encourages researchers, practitioners and family stakeholders to recognize the value of the child as an active agent in shaping the health promoting potential of their family context.

## Introduction

### Understanding the importance of the family as a setting for health promotion

The objective of this scoping review is to provide details as to how conceptual and theoretical models of the health promoting potential of the family are being used in health promotion contexts. This knowledge is important because the family is a key setting for health promotion. Throughout infancy and childhood, we live among others who can provide for our basic needs, guide and nurture us as individuals, and launch us on health trajectories that follow us throughout the life course. Socioecological models place individuals within families and depict family settings as the most intimate context of health and social influence [1, 2].

### Why are models of "health promoting settings" important?

Health promotion practitioners often leverage the structure that exists in the physical and social environments of the settings in which everyday life unfolds in order to establish scaffolds for programs and services. The health promoting school, for example, has developed as a well-articulated context where healthy policy, health education, health environmental features and partnerships can be established [3–5]. Similarly, other health promoting environments have been described in detail, including health promoting outdoor environments [6], health promoting workspaces [7], health promoting hospitals [8] and health promoting municipalities [9].

### The health promoting family–a conceptual framework

In 2004, Christensen added to this dialogue by proposing a conceptual model of the "health promoting family" [10]. In doing so, she drew attention to the scarcity of research related to how families engage in promoting their health in the context of their everyday lives and argued for the importance of increased understanding about how the family can play a part in promoting both the health of children and the children's' capacities as health-promoting actors. Along with environmental factors such as income, education and resources, she suggested an emphasis on the family's ecocultural pathway (family values and goals) and family practices (including practices around food, physical activity, risk behaviors and meaningful social connections) for promoting health. In addition to adult or parental figures in families, core to Christensen's model is the importance of the child as a "health promoting actor" who has opportunity to participate in, contribute to, and manage their own health and well-being [10].

As we engaged with Christensen's model [10], we were struck by how underdeveloped conceptual and/or theoretical frameworks of health promoting families appeared to be in comparison to frameworks that have been developed to describe and guide other settings. Indeed, while the family is repeatedly noted as an essential and universally critical context for health promotion, the development of conceptual modeling for a "health promoting family" is

limited. We also noted how limited any attempts in the literature have been to clearly define what might constitute a "health promoting family." To date, such a definition does not appear to exist. There are numerous likely reasons for these gaps, including that family, parenting and child development are intimate and culturally bound activities which vary significantly across homes and settings and for which authority remains largely in the personal versus the public, state or organizational sphere. Further, families are complex and diverse. Any attempt to delineate what might characterize a family as a health promoting context must be broad and flexible enough to recognize the complexities of real people's lives. Indeed, some research has moved from setting up a false normal of what a family should look like, to a focus on what families do, and how they operate as a unit [11–14].

Prompted by our examination of Christensen's model, we conducted a scoping review with the objective of identifying, analyzing and interpreting conceptual and theoretical frameworks or models that focus on the health promoting potential of the family context. A scoping review was appropriate in that it enabled us to conduct a broad, interdisciplinary survey of previous research with the purpose of identifying key characteristics related to the concept of the health promoting family [15]. Our hope was that we would be able to use the findings from this review to inform research on family health by building on current and high-quality evidence. Further, we anticipated that this synthesis of knowledge would be valuable to practitioners who are involved in health promotion and whose work involves supporting families in their own contexts. Finally, through this review, we hoped to identify strengths and gaps in the ways that health promoting families are modelled in the academic literature and inform future initiatives at such modelling.

## Methodology

### Overview

The approach to this scoping review was adapted from the PRISMA [16] guidelines for scoping reviews. Guidance in formulating our search strategy was sought from a Senior Health Sciences Librarian at the Bracken Library at Queen's University, Kingston, Ontario.

A three-step search strategy was used to find relevant papers in order to contribute to answering the question: How is the health promoting potential of the family portrayed in conceptual and theoretical models in academic and grey literature? In step one, studies were identified by key-word searching electronic databases: Medline (1996–2021); PsycINFO (1967–2021); Embase (1996–2021); and CINAHL (1981–2021). For example, we used the following search strategy in Ovid MEDLINE(R) without revisions (<1996 to Present-June, week 2, 2015) and (June week 2, 2015 –Present-September, 2020) was: ((family [MeSH terms] OR family characteristics [MeSH terms] OR family relations [MeSH terms] OR parent-child relations [MeSH terms] OR nuclear family [MeSH terms]) OR family health [MeSH terms]) AND ((models, theoretical [MeSH terms] OR models, educational [MeSH terms]) OR conceptual framework$.[abstracts and titles] OR conceptual model$.[abstracts and titles] OR theoretical framework$.[abstracts and titles] OR theoretical model$.[abstracts and titles]) AND (Health Behaviour [MeSH terms] OR Health Promotion [MeSH terms] OR Health Knowledge, Attitudes, Practice [MeSH terms] OR health status [MeSH terms] OR Nutritional Status [MeSH terms] OR exp. obesity [explode, MeSH terms] OR "Social Determinants of Health" [MeSH Terms] OR exp. social environment [explode, MeSH terms] OR support$.[abstracts and titles] OR strong famil$.[abstracts and titles]). Fig 1 describes the search string that was adapted for each database.

Step two involved a hand search of the archives of the Journal of Marriage and Family, a search of the reference lists of included studies, and a thorough backward and forward search

```
1   family/ or family characteristics/ or family relations/ or
    parent-child relations/ or nuclear family/
2   Family Health/
3   1 or 2
4   models, theoretical/ or models, educational/
5   conceptual framework$.ab.ti.
6   conceptual model$.ab.ti.
7   theoretical framework$.ab.ti.
8   theoretical model$.ab.ti.
9   Health Behavior/
10  Health Promotion/
11  Health Knowledge, Attitudes, Practice/
12  health status/
13  Nutritional Status/
14  exp Obesity/
15  "Social Determinants of Health"/
16  exp social environment/
17  support$.ab.ti.
18  strong famil$.ab.ti.
19  9 or 10 or 11 or 12 or 13 or 14 or 15 or 16 or 17 or 18
20  4 or 5 or 6 or 7 or 8
21  3 and 19 and 20
22  limit 21 to English language
```

**Fig 1. Search string.**

using Google Scholar and Web of Science for Christensen's key article [10], each of which enabled us to identify additional studies. In step three, we conducted an intentional search for grey literature that may not have been found in the scientific databases that we searched in steps one and two. This step generated an additional set of models from textbooks, dissertations, thesis manuscripts, literature reviews, academic journals and reports.

English language documents that included an illustrated model related to the concept of the health-promoting family were included. Sources were excluded if they did not mention families that included adult(s) and child(ren) or if the outcomes or exposures of interest were not related to individual or family health. No additional restrictions were set on study date, study design, types of families, types of exposures or outcomes. After duplicates were removed, titles were reviewed by a research assistant to exclude articles that obviously did not meet inclusion criteria. All abstracts and then full text articles were reviewed by VM and either CD (studies up until 2017) or KP (studies from 2017 to 2020). A data charting spreadsheet was jointly developed by VM, CM and KP to determine which models to include. Three researchers (VM, CM, and later KP) independently charted the data, discussed results and updated the spreadsheet through an iterative process as inclusion and exclusion decisions were made. This project spanned multiple years. The first stage involved a search for models between the earliest date possible for each database up to June (week 2, 2015) that took place between June and August 2015. The second stage involved a search for models between June (week 2, 2015) and September, 2020. A research assistant (JB) was involved with every aspect of this scoping review until 2017. A postdoctoral fellow (KP) then provided extensive input in all aspects of this literature scan throughout 2020. To synthesize our results, we initially grouped the models by the disciplines from which they emerged and the family characteristics that were identified. As we engaged in an iterative and inductive process of analysis and critical discussion between researchers, we identified further ways of synthesizing the models. This included synthesizing the ecological and environmental factors that were identified as important; the health promoting features of the family; and the role of the child as an active or passive agent in promoting family health.

# Results

## Study selection

After applying the inclusion and exclusion criteria, the overall search from all three steps generated 113 included manuscripts/chapters with 118 unique models relevant to the "health promoting family". The flow diagram depicted in Fig 2 outlines the steps that we used to arrive at the included studies and unique models in our search results.

## Summary table of identified models

Table 1 provides a summary of the 118 distinct models that our review yielded. It includes: (1) the name of the model (including variations on the model that are included in the same source); (2) a short description of each model; (3) a description of the child's role in shaping

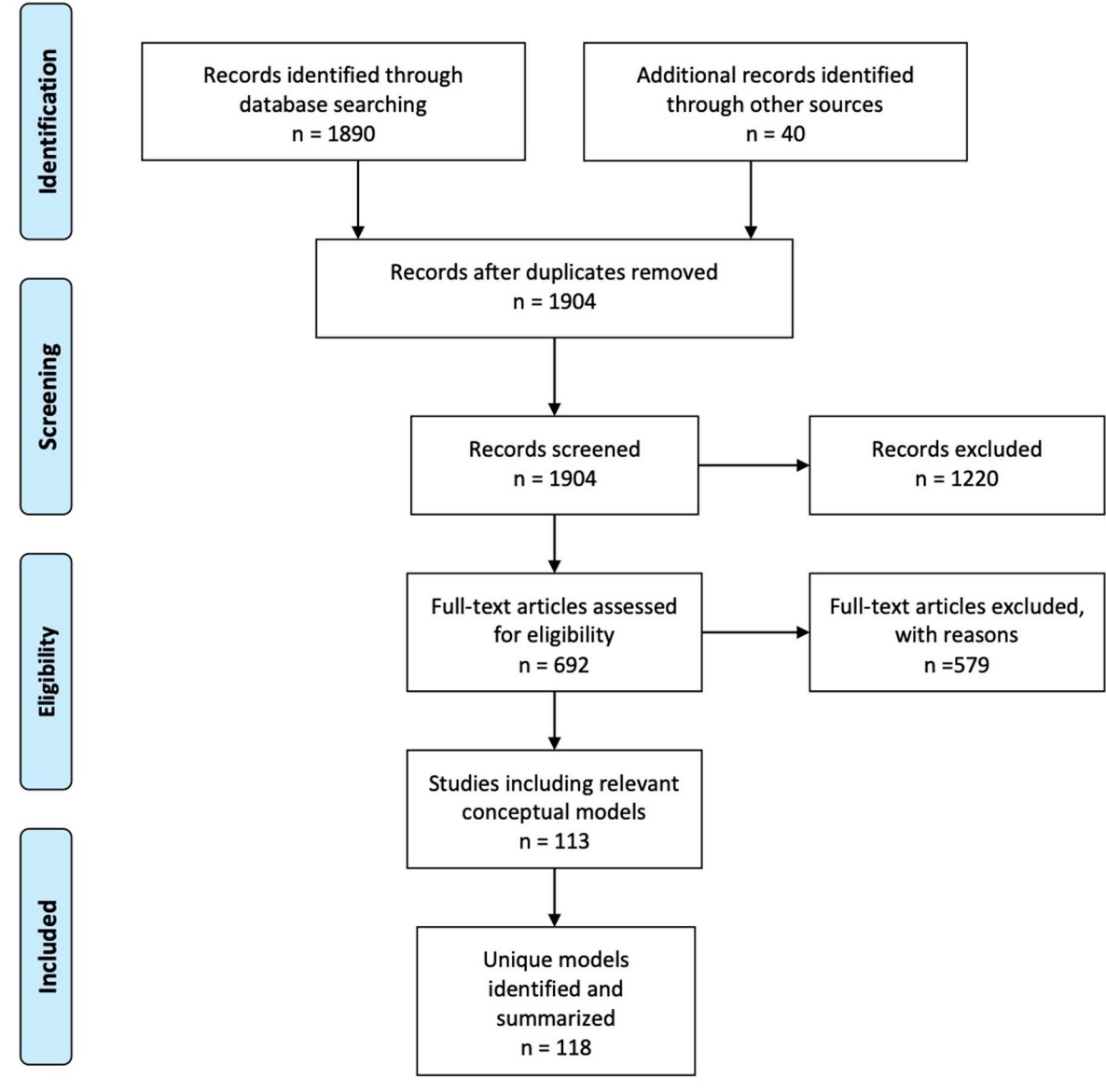

**Fig 2. Flow diagram of included studies.**

**Table 1. Summary table of identified models.**

| # | Model name | Model description | Child's Role | Source (See references for full citation) |
|---|---|---|---|---|
| 1 | Figure 8.3. Supportive factors and mothers' agency in the school environment. | This model details supportive factors at school that empower mothers' in Saudi Arabia roles in their child's oral health at home. Extension between school and home, sustainability of oral health programs, obligation of children and parents for engaging in oral health activities, and authority between mothers and teachers influence on children's daily oral health behaviors can support mother's agency related to their children's oral health. | Passive | Aldossari, 2016 [17]. |
| 2 | Figure 2. Conceptual framework on malnutrition. | This model considers how household characteristics, maternal characteristics and other structural factors (region, urban/rural) influence child malnutrition. | Passive | Annim, Awusabo-Asare, & Amo-Adjei, 2015 [18]. |
| 3 | Figure 1. Conceptual model. | This model describes pathways between key environmental and social stressors, parental characteristics and child characteristics that affect family functioning and child resilience. It describes social supports as mediators between environmental and personal stressors and outcomes. | Passive | Armstrong, Birnie-Lefcovitch, & Ungar, 2005 [19]. |
| 4 | Figure 1. Model linking income, material hardship, and parenting to child health status. Figure 2. SEM. | This causal model illustrates multiple mediating pathways of income-poverty, material hardship (food insufficiency, medical need), parental depression, positive parenting behaviors, and child health status. | Passive | Ashiabi & O'Neal, 2007 [20]. |
| 5 | Figure 1. Conceptual model for SOL youth: Understanding risk and protective factors for Latino childhood obesity. | This model describes risk and protective factors for childhood obesity on the individual, interpersonal, organizational, and community level, and emphasizes the impact of families and parenting on obesity risk. | Active/ Passive | Ayala et al., 2014 [21]. |
| 6 | Figure 1. Expected associations among family communication orientations, health-specific conversation factors, and health outcomes. Figure 2 Multilevel structural equation model (SEM). | This model examines how aspects of family communication promote or prevent health attitudes and behaviors (specifically, physical activity and diet). It examines communication behaviors as they relate to health outcomes. | Active | Baiocchi-Wagner & Talley, 2013 [22]. |
| 7 | Figure 2. Potential mediating role of father in relationships linking health determinants to child health; Figure 3. Research evidence of direct effects of father's involvement on child development; Figure 4. Research evidence of direct effects of father's absence on child development; Figure 6. Reciprocally causal links among health determinants and outcomes of father's involvement. | This model demonstrates the role that fathers play in shaping child health outcomes. Figure 2 illustrates the potential mediating role of fathers between determinants of health, the family environment and child health outcomes. Figure 3 provides a schematic organization of positive effects gained from the involvement of fathers on four dimensions of child health outcomes (cognitive, academic, psychological/ emotional and social). Figure 4 offers a model of direct effects of a father's absence on child development and health outcomes (including adaptive functioning, academic, psychological/emotional and social). Figure 6 shows the relationship between direct and indirect impacts of father engagement in parenting and determinants of health. | Passive | Ball, Moselle, & Pedersen, 2007 [23]. |
| 8 | Fig. 1. Final two-factor oblique confirmatory factor analysis model of family entropy with standardized parameter estimates. | This model examines how family entropy in the home environment (household organization and household disorganization) is related to child health. Household organization can help prevent child obesity and household disorganization can have detrimental child health effects. | Passive | Bates et al., 2019 [24]. |
| 9 | Figure 1. An integrated model of social environment and social context. | This model seeks to describe the nature of the relationship between the social environment and social context including how people, place, activity, objects and time are all aspects that influence child experiences. | Active | Batorowicz et al., 2016 [25]. |
| 10 | Figure 1. Theoretical cascade model linking provider delivery, participant responsiveness, and improvements in program outcomes. | This model depicts how dimensions of provider delivery influences program outcomes through participant responsiveness. Positive engagement by facilitators can positively influence parent attendance and competence in home practice. | Passive | Berkel et al., 2018 [26]. |

*(Continued)*

**Table 1.** (Continued)

| # | Model name | Model description | Child's Role | Source (See references for full citation) |
|---|---|---|---|---|
| 11 | Figure 2–3. Model of community nutrition environment (Glanz, Sallis, Saelens, & Frank, 2005). | This model describes the policy, environmental, and individual variables that relate to eating patterns in children. This model associates environmental influences as mediators of eating behaviors of children. | Passive | Bertrand, 2019 [27]. |
| 12 | Figure 2–4. Ecological framework depicting the multiple influences on what people eat (Story, Kaphingst, Robinson-O'Brien, & Glanz, 2008). | This model seeks to understand the macro-level, physical environments, social environments and individual factors that impact dietary behaviors of school children. This model offers an ecological framework detailing the various influences on children's eating behaviors. | Active/ passive | Bertrand, 2019 [27]. |
| 13 | Figure 1–1. Realms of family life: A focus of family health nursing practice. | This framework details a number of family processes that contribute to aspects of family well-being, including family coping processes, interactive processes, integrity processes, developmental processes, and health maintenance. It also acknowledges the reciprocal impact that family members have on each other. It is used to show that a collaborative family nursing process must include assessment of multiple aspects of family life. | Active | Bomar, 2004 [28]. |
| 14 | Figure 1. Adaptation phase of the resiliency model of family stress, adjustment, and adaptation. | This model illustrates the process of family adjustment and adaptation. When challenge is experienced, a family enters the adaptation phase, in which functioning determines the outcome of adaptation. Balance and harmony within the family can promote problem solving and coping, as do family resources (social support) and situation appraisal. Overall, it examines the factors that enable families with children to adapt to stress. | Active | Brown, Fouche, & Coetzee, 2010 [29]. |
| 15 | Figure 2. The ENERGY- project specific ENRG (Environmental Research for weight Gain prevention) framework. | This model focuses on how aspects of the family and school environments can influence energy-balance choices (dietary, physical activity, sedentary, sleep) for weight-gain prevention. Aspects of the family environment include parental rules, feeding style, and parent's BMI. Aspects of the school environment include availability of healthy and unhealthy food options, school food policy and physical activity opportunities. | Active | Brug et al., 2010 [30]. |
| 16 | Fig.1 Conceptual model of social determinants of health and racial/ethnic disparities in T2DM. (Adapted with permission from: Walker RJ et al. BMC Endocr Disord. 2014;14:82; with permission from BioMed Central) | This model illustrates how social determinants of health including low family income, low levels of parental educational attainment, and higher stress in youth can indirectly contribute to onset Type 2 diabetes and consequent adverse psychosocial outcomes. It suggests that these social determinants can influence health behaviors, health knowledge, coping/problem solving and ultimately, diabetes outcomes. | Active | Butler, 2017 [31]. |
| 17 | Figure 1. Conceptual Model 1 with caregiver-adolescent discrepancies. Figure 2. Conceptual Model 2 with adolescent and caregiver acculturation main effects. | Both of these conceptual models seek to examine how acculturation-related variables impact adolescent health risk behaviors and depressive symptoms (HRB/DS) as mediated by caregiver and adolescent reports of family functioning. Conceptual Model 1 examines discrepancies between caregiver-adolescent acculturation and Conceptual Model 2 examines the individual effects of caregiver and adolescent acculturation components. | Active | Cano et al., 2016 [32]. |
| 18 | Figure 3.5. Proposed integrated conceptual model for the understanding pathways that influence child development and the impact of child health on the family. | This model uses a life-course perspective to describe the pathways that influence child development, and the impact of child health on the family. It recognizes that different factors affect the child and family at different stages of life. | Passive | Cheng, 2013 [33]. |
| 19 | Figure 1. Bruhn and Parcel model of health promotion (1982). | This model details how family influence and adolescent development characteristics influence adolescent health behavior and health status. Components of family influence on adolescent health behaviors include reinforcement, modeling, interaction patterns, constraints & opportunities. | Active | Chiu, 2005 [34]. |

(*Continued*)

**Table 1.** (Continued)

| # | Model name | Model description | Child's Role | Source (See references for full citation) |
|---|---|---|---|---|
| 20 | Figure.1. Conceptual framework based on the theory of planned behavior: Factors that influence children's beverage consumption behaviors. | This conceptual framework uses the theory of planned behavior to understand how attitude, perceived behavioral control and subjective norms of parent factors, child factors and external factors influence parental behavioral intention which relates to child's beverage consumption. Parent factors include sociodemographic and parental health/behaviors. Child factors include demographic, taste preferences, health status, temperament and modeling. External factors include family interactions, beverage availability, household beverage rules, physician/nutritionist, neighborhood, media, policies, and early childcare programs/daycare. | Active/ passive | Choy & Isong, 2018 [35]. |
| 21 | Figure 1. Model of the health-promoting family. | This model of the health-promoting family illustrates how external influences on the family (community and societal), and well as processes internal to the family (family ecocultural pathway, genetics/ family health history, health practices) shape child health status. The child is viewed as a health-promoting actor, and the degree to which children act in ways so as to promote (or demote) their own health is considered to be an important aspect of family life. | Active | Christensen, 2004 [10]. |
| 22 | Figure 1. Summary of themes and subthemes identified in qualitative interviews. | This model illustrates how parents' motives for participation in physical activity influence provision of PA for their children with a visual impairment. It explains that while parents are committed to providing PA for their children with VI, they encounter challenges in engaging their children in PA, including: the impact of visual impairment on PA, parental teaching proficiency of PA, inadequacies in PA programming and influences of others' opinions. Access to functional support systems can influence parental motivation in the provision of PA for their children with a visual impairment. | Passive | Columna et al., 2019 [36]. |
| 23 | Figure 2.1. Child-parent reciprocal influences model. | This model proposes family, individual, and interpersonal factors that influence health promoting behaviors. | Active | Coviak, 1998 [37]. |
| 24 | Figure 5.1. The model of concept of well-being in older Taiwanese. | This two-part model uses the circle to represent wholeness and happiness in the concept of well-being. It suggests that five basics of well-being (family support, extra familial support, completion of family obligation, sense of dignity and self-reliance) are essential to comprehensive well-being of self and family. | Passive | Dai, 1995 [38]. |
| 25 | Figure 5.1. A graphical depiction of the impact of family processes on children's emotional insecurity in the family and their trajectories of adjustment within the reformulated emotional security theory. | This model describes how parenting practices and conflict impact child emotion, development and adjustment, and consequently influence child health overall. | Passive | Davies, Sturge-Apple, & Martin, 2013 [39]. |
| 26 | Figure 2. The revised family ecological model; bolded text and boxes indicate new components and constructs that were not part of the original model. | This model is a revised version of the Family Ecological Model. It illustrates a causal sequence whereby family ecology, family social/emotional context, and parenting practices influence family health outcomes. The focus of this model is on family and child obesity prevention. Both family environment and family/child health outcomes are detailed. | Active | Davison, Jurkowski, & Lawson, 2013 [40]. |
| 27 | Figure 1. Structural model linking mothers' gendered roles and ideologies to adolescent depression. Figure 2 (SEM modelling). | Figure 1 demonstrates how congruity/incongruity between the mother's actual role and the mother's acceptance of ideologies around traditional gender roles can relate to adolescent depression. Figure 2 uses the same variables as Figure 1, but depicts the unstandardized parameter estimates and standard errors for all significant paths. Insignificant paths were not deleted from the statistical model but are not depicted in the diagram for simplicity of presentation. | Passive | De Coster & Zito, 2013 [41]. |

(*Continued*)

**Table 1.** (Continued)

| # | Model name | Model description | Child's Role | Source (See references for full citation) |
|---|---|---|---|---|
| 28 | Fig. 1. Social-ecological model applied to the Kanyakla Nutrition Program (Gregson et al., 2001; Stokols, 1992). | This model illustrates the intersection between the social-ecological model and the Kanyakla Nutrition Program (p. 197). The intersection between individual, interpersonal, institutional and organizational, community, and structures and systems can influence the uptake and effectiveness of the Kanyakla Nutrition Program. | Passive | DeLorme et al., 2018 [42]. |
| 29 | Fig 2. Conceptual model of the Kanyakla nutrition program. | This model represents how the intervention actions in a nutrition program and the effects of the community health workers engagement influence community effects and household/child outcomes, including maternal and child nutrition behavior, household food security, improved maternal and child nutrition. | Active/ passive | DeLorme et al., 2018 [42]. |
| 30 | Figure 1–1. Social construction of family health. | This conceptual model illustrates contextual (internal [in the household] and external [social, historical, political]), structural (family health routines and health behaviors) and functional (individual and family processes) aspects of the social construction of family health. | Passive | Denham, 2003 [43]. |
| 31 | Figure 9–1. Social construction of family health definitions and practices. | This model illustrates how family environment and relationships influence family perceptions of health and engagement in patterns of health behavior (normative [health promoting] and non-normative [health depleting]). It demonstrates how lived experiences lead to beliefs, values, behaviors that influence our health decisions. | Passive | Denham, 2003 [43]. |
| 32 | Figure 12–1. Factors affecting the modification of the family health constructs. | This model describes a number of categories that influence and modify the family health construction, including parental beliefs and values, temporal patterns, ecological context, accommodation of unpredictable events, relational interactions, and knowledge exposure. | Passive | Denham, 2003 [43]. |
| 33 | Figure. Theoretical model adopted in the study. | This model outlines the influence that gender, SES, home environment and subjective aspects related to parental perceptions about oral health and children's own perceptions about self-oral health has on dental caries in school children. | Active/ passive | dePaula et al., 2015 [44]. |
| 34 | Figure 11.1. Family formation in low-income populations. | This model details indirect and direct influences on child well-being, with a focus on family-level factors. Child well-being is influenced by parenting characteristics and behavior as well as family relationships and functioning. The model also considers the impact distal influences (such as culture, policy, and economics) have on the family and parent functioning. | Passive | Dion et al., 2003 [45]. |
| 35 | Fig. 1. The relationship between parent-child, stimulation and dental caries: a life course approach. | This model proposes that parent behavior is related to social exposure in the child's first years of life, increasing the risk of chronic diseases like dental caries. Parental characteristics and determinants of health can negatively influence child caries and this can negatively influence risk of health outcomes in the child. | Passive | dos Santos Costa et al., 2019 [46]. |
| 36 | Figure 1. Conceptual model of influences on adherence to paediatric asthma treatment. | This comprehensive model describes the role of family functioning and child and parent psychological factors in adherence to paediatric asthma treatment. | Passive | Drotar, & Bonner, 2009 [47]. |
| 37 | Figure 1. Final structural equation model relating latent constructs of neglect to children's functioning. | This model is used to quantitatively assess relationships between child functioning (measured by externalizing and internalizing behavior and social problems) and various family constructs. It demonstrates that parental support, affection, and family conflict all predicted children's later functioning. | Passive | Dubowitz et al., 2005 [48]. |

(*Continued*)

**Table 1.** (Continued)

| # | Model name | Model description | Child's Role | Source (See references for full citation) |
|---|---|---|---|---|
| 38 | Figure 1. Schematic illustration of the theoretical model. | This figure models plausible pathways between community and family variables and individual determinants of childhood dental caries illustrated in the Fisher-Owens conceptual model. | Passive | Duijster et al., 2014 [49]. |
| 39 | Figure 1. Theoretical model of chaos and child health. | This model explores chaos in the household and family system as a determinant of child health. Chaos in the household (physical and social disorder) can result in environmental exposures and stress/ lack of emotional support, which can result in reduced child health. Similarly, work/ family child-care chaos can lead to maternal stress, lack of illness care and supervision, and decreased child health. The model acknowledges that poor child health also feedbacks into more chaos, exacerbating the cycle. | Passive | Dush, Schmeer, & Taylor, 2013 [50]. |
| 40 | Figure 1. Conceptual model; Figure 2. Model for female and male younger siblings (SEM); Figure 3. Model for female younger siblings (SEM)]. | Figure 1 provides a conceptual model that suggests that aspects of mothers' parenting and specific stresses within the family shape relationships between youths' siblings. In turn, these relationships influence adolescents' drug use, high-risk sexual behaviors and associated outcomes (i.e. pregnancy and sexually transmitted disease). Figure 2 demonstrates that certain qualities of the sibling relationship (i.e. high older sister power, low warmth/closeness) served as mediators between the risk behaviors of younger siblings. Model estimates for girls only are shown in Figure 3. | Passive | East & Khoo, 2005 [51]. |
| 41 | Figure 2–1. Conceptual framework for reviewing 'obesogenic landscapes' in urban children's geographies. | This model seeks to illustrate the factors that influence independent outdoor play in urban children's geographic locations. This model represents the cultural and socio-economic characteristics of children and their home environment that play a role in the participation of diverse outdoor activities for urban children. Activities can be influenced and vary by seasonality, city, neighbourhood, home, if transportation is needed, play space, urban design and safety. | Active/ passive | Ergler, 2012 [52]. |
| 42 | Fig. 1. Thematic analysis. | This model represents children's food related health literacy practices. It describes how children access health information and sources of information related to food and healthiness. This model also describes how children understand health information and give meaning to healthy and unhealthy perceptions of this information. | Active | Fairbrother et al., 2016 [53]. |
| 43 | Figure 2. Modified model predicting family adaptation. | This model shows linkages between stress (objective and perceived), family resources, and coping strategies and how these influence family adaptations to living in a war zone. Family resources supported family adaptation, and coping strategies partially supported adaptation. | Passive | Farhood, 1999 [54]. |
| 44 | Figure 1. Child, family, and community influences on oral health outcomes of children. | This model takes a holistic approach to examine how individual, family, and community influence oral health outcomes in children. It presents a number of detailed community, family, and child level influences child oral health, and recognizes the role of time and environment on oral health outcomes. On the family level, both physical (physical safety, family composition, etc.) and relational (family functioning, social support, etc.) factors are acknowledged to influence child oral health. The child is recognized as a potential health-promoting actor. | Active/ Passive | Fisher-Owens et al., 2007 [55]. |

(*Continued*)

**Table 1.** (Continued)

| # | Model name | Model description | Child's Role | Source (See references for full citation) |
|---|---|---|---|---|
| 45 | Figure 1.2. Pathways of family processes; Figure 1.3. Life process of the family system. | This framework illustrates healthy family processes that result in family congruence (harmony, compatibility). In accomplishing family tasks and striving towards the targets of stability, growth, control, and spirituality, congruence is the goal. Family tasks (ranging from physical care, emotional support, reproduction, culture maintenance, family commitment, acceptance, enhancement of social skills, etc.) occur within the earthly influences of space, time, energy, and matter. | Passive | Friedemann, 1995 [56]. |
| 46 | Figure 1. The PEN-3 cultural model. | This model posits that cultural factors influence African-American mothers' and their daughters' HPV vaccine acceptance. In the PEN-3 cultural model culture (cultural identity) is a key facilitating or deterring factor in preventive health behaviors. Relationships and expectations including perceptions, enablers and nurturers can influence performing a health behavior. | Passive | Galbraith-Gyan et al., 2019 [57]. |
| 47 | Figure 1. Familial approach to the treatment of childhood obesity: conceptual model. | This model outlines how parents can influence the attainment of healthy weight in children by modeling a healthy lifestyle, changing the home environment, and by promoting health habits in children. Through parental cognitive and behavioral change (increased nutrition & health skills and increased parenting skills) and environmental change (healthy environment in family/ home), parents can help their children to attain a healthy weight status. | Passive | Golan & Weizman, 2001 [58]. |
| 48 | Figure 1. Adapted from the transactional stress and coping (TSC) model of adjustment to chronic illness (Thompson et al. 1994) for siblings. | This model hypothesizes that child well-being and adjustment (to sibling illness) will be a function of relationships between the ecological, family, and sibling adaptation process. Family variables such as extended family, family functioning, family coping, sibling coping and efficacy impact the outcome (adaptation and well-being) on the ecological level and also mediate the association between illness and adaptation. | Active | Gold et al., 2008 [59]. |
| 49 | Fig. 1. FRESH theoretical model. | This model illustrates the intervention components that influence family participation in physical activity (PA). The FRESH intervention components are based on FRESH, a goal setting and self-monitoring intervention to increase PA in families where families choose new weekly challenges with their children as the leads and receive rewards for completing them. These components are cyclical and have an impact on outcomes of screen-time, quality of life/wellbeing, family functioning, physical health, all of which influence physical activity. | Active | Guagliano et al., 2019a [60]. |
| 50 | Figure 3. FRESH theoretical model. FRESH, Families Reporting Every Step to Health. | This model illustrates a FRESH logic model that details the intervention components and the family and individual-level mediators that influence outcomes of physical activity and screen time behavior and ultimately, health and wellbeing. | Active/ passive | Guagliano et al., 2019b [61]. |
| 51 | Fig.1 A conceptual model of influence of family dynamics and sleep health behaviors on hypertension risk. | This model illustrates the relationship between family dynamics (relationship quality, conflict, shared health behaviors), sleep health behaviors (sleep duration, timing and quality) and hypertension risk in children and youth. Family dynamics are associated with hypertension risk and family dynamics combined with youth sleep health behaviors are associated with hypertension risk in children and youth. | Passive | Gunn & Eberhardt, 2019 [62]. |
| 52 | Figure 1. | This model demonstrates the relationship between perceived prolonged parental grief and current functioning as they relate to child perceptions of interparental conflict (CPIC). Perceived prolonged parental grief is a predictor of emotional security preoccupation which in turn, is a predictor of current functioning. | Passive | Hardt et al., 2019 [63]. |

(*Continued*)

**Table 1.** (Continued)

| # | Model name | Model description | Child's Role | Source (See references for full citation) |
|---|---|---|---|---|
| 53 | Figure 1. Conceptual model for predictors of children's development. | This model shows how child development is influenced by child, parental, and family factors. Individual child and parent characteristics influence family climate and relationships and child-self regulatory process. These relationships/ family process and child traits consequently influence child development and behaviors. | Active | Hauser-Cram et al., 2001 [64]. |
| 54 | Figure 2. Conceptual model for predictors of parent well-being. | This conceptual model suggests that parent well-being is influenced by child and parent related stress. Both parent (education, marital status, assets) and child (age, sex, disability) individual characteristics influence family climate and relationships within the family, in addition to child skills. All of these consequently affect parent well-being. | Passive | Hauser-Cram et al., 2001 [64]. |
| 55 | Figure 1. Conceptual Model of the Influence of Macro- and Family-Level Sociocultural Contextual Factors in Youth and Pubertal Timing on Women's Lifetime Educational Achievement. | This conceptual model seeks to examine both life events and sociocultural contextual factors in youth that have an impact on lifetime educational achievement. Life events and sociocultural contextual factors occur at the macro and family levels. | Passive | Hendrick et al., 2016 [65]. |
| 56 | Figure 1. Obesity resistance model: a summary of the interactions of family environmental factors influencing children's weight status and behaviors. | This model quantitatively summarizes the interactions between parent, family and child factors that influence child weight status and related behaviors. Parent health behaviors and knowledge impact their parenting style and feeding practices in addition to family environment (food and physical activity). Child screen time, exercise, BMI, and fruit and vegetable intake are influenced by parent behavior and family environment. | Active/ Passive | Hendrie, Coveney & Cox, 2012 [66]. |
| 57 | Figure 1. Relationship-based feeding framework. | This model provides a framework detailing the child–caregiver relationship and the biopsychosocial and contextual factors that affect the feeding relationship, and it promotes active engagement on the part of the child and caregiver within the context of family relationships, community support, and resources. | Active | Henton, 2018 [67]. |
| 58 | Figure 1. Model illustrating the mediation paths for the combined sample (top panel), ASD+parent-reported below average IQ (middle panel), and ASD+parent-reported average or above IQ (bottom panel) between ASD severity, parental romantic expectations for their child, and number of sex- related topics covered by parent. | This model compares the parents of youth with ASD + parent-reported below average IQ and average to above average IQ in relation to parental provision of sexuality and relationship education via ASD symptom severity. Parental romantic expectations are influenced by above or below average IQ of youth with ASD. | Passive | Holmes et al., 2016 [68]. |
| | Figure 1. Conceptual framework of life events and cultural processes that shape maternal capabilities and influence child nutrition and hygiene care behaviors. | This model illustrates how life events influence maternal capabilities which then influence a mother's capabilities. This impacts child nutrition and hygiene. Cultural events and processes shape maternal capabilities, which can negatively impact caring for children. | Passive | Ickes et al., 2017 [69]. |
| 59 | Fig. 1. Social ecological model applied to child health (Kazak, 2006). Fig. 2. Coding tree, based on the social ecological model applied to child health by Kazak (2006). | These models illustrate how social ecological factors can influence a child's health and quality of life in those children with paediatric illnesses. These factors are part of interactive systems that can explain barriers and facilitators of social functioning of children with paediatric illness that influence child health. The systems include microsystem (child, parents, siblings, family, illness), exosystem (hospitals, school, peers, neighborhoods, social network), and macrosystem (cultures, religion, law, social class and technology). | Active/ passive | Janin et al., 2018 [70]. |

(*Continued*)

**Table 1.** (Continued)

| # | Model name | Model description | Child's Role | Source (See references for full citation) |
|---|---|---|---|---|
| 60 | Figure 1. Empirical model that summarizes the study's findings, based on the adolescents' voices and the researchers' interpretation of the empirical data through self-determination theory (SDT). | This model highlights facilitators of physical activity for adolescents. In order for adolescents to engage in physical activity, it must be fun and enjoyable. Enjoyment is impacted by variation, physical skills and friends. Enjoyment from physical activity that is supported by family and friends and a supportive school environment helps to foster a sense of autonomy and competence which facilitates engagement in physical activities. | Active | Jonsson et al., 2017 [71]. |
| 61 | Figure 1. "Influence of child, family, and community on oral health outcomes of children" (Fisher-Owen et al., 2007). | This multi-level model outlines five domains that influence child oral health outcomes. It includes community level, family-level and child-level influences, which are bound by time and environment, and shape children's oral health. | Active/ passive | Kalil, 2017 [72]. |
| 62 | Figure 1. Proposed mediation model; Figure. 2. Direct model without mediation; Figure. 3. Full mediation model. | This model shows how parent health-related feeding goals and feeding practices influence child-eating behaviors in positive and negative ways. | Active/ Passive | Kiefner-Burmeister et al., 2014 [73]. |
| 63 | Figure 1. A unifying conceptual model for early childhood caries (ECC) showing the connections between social, environmental, maternal, and child factors. | This model illustrates how a number of environmental, family (especially maternal) and child factors influence early childhood caries. Maternal characteristics are influenced by economic and family stress and environment/ social disadvantage; consequently, these maternal factors influence child dental behaviors and outcomes. Parenting is also affected by maternal and family stress, which has important implications for child dental behaviors. | Passive | Kim Seow, 2012 [74]. |
| 64 | Box 9–1. Characteristics of Healthy Family. | This table/ model outlines the characteristics of healthy family. There are 6 main domains within 3 categories that contribute to family health. Unity is marked by commitment and time together; flexibility is measured by family ability to deal with stress and spiritual well-being. Family communication is broken down into positive communication and appreciation and affection. | Passive | Kim-Godwin & Bomar, 2010 [75]. |
| 65 | Figure 1. A model of factors affecting the participation of children with disabilities. | This is a conceptual model of the environmental, family, and child factors thought to influence child participation in recreation and leisure activities. It examines the interaction between environmental, family, and child factors that influence a disabled child's participation in activities (daily, recreational, physical, etc.). Family demographics, financial and time impact, environment, and preferences for recreation are impacted by environmental factors and consequently influence child factors. | Active | King et al., 2003 [76]. |
| 66 | Figure 1. Family systems theory framework related to youth health behaviors. | This framework explores how the family system may influence health behaviors in youth. It demonstrates that positive parenting styles are associated in improvements in youth health behaviors, including physical activity, weight loss and diet. | Passive | Kitzman-Ulrich et al., 2010 [77]. |
| 67 | Figure 2. Model for family psychosocial well-being in a South African context. | This model illustrates the dimensions of family psychosocial well-being. The main interactive relationships are between family, family strengths, and family functioning. Interactions within family and family psychosocial well-being can influence or be influenced positively or negatively by other internal and external factors (family interactions, values, support, etc.). The model also acknowledges external influence of healthy friendships, education, communities, and safe environments. | Passive | Koen, van Eeden, & Rothmann, 2013 [78]. |

(*Continued*)

**Table 1.** (*Continued*)

| # | Model name | Model description | Child's Role | Source (See references for full citation) |
|---|---|---|---|---|
| 68 | Figure 1. Conceptual framework explaining the relationship between family structure, number of siblings and child well-being. | This model explains the relationship between family structure, number of siblings and child well-being. Based on Albrecht et al. (1994), it shows how family structure and number of siblings are associated with presence and distribution of family resources, in addition to care/decision making of adults. | Passive | Kumar & Ram, 2013 [79]. |
| 69 | Figure 1. Theory of change of M-PACT+ | This model represents the steps of change in the theory of change that was identified through data collection. These steps of change include a top down approach where facilitators and schools identify and introduce families to M-PACT +. Changes occur after participation by families. | Active | Laing et al., 2019 [80]. |
| 70 | Figure 2.3. Conceptual framework for the development of nurse-led health promotion visiting programme and family health. | This models offers a framework that summarizes the relationship between programme factors that enhance family health promotion initiatives that improve healthy behaviors and family health in relation to seasonal influenza. Predisposing, reinforcing and enabling factors help determine delivery of nurse-led visiting programs that influence family health promotion initiatives in child health and in turn improve family health and health behavior maintenance. | Passive | Lam, 2016 [81]. |
| 71 | Figure 1. Conceptual model of how parents influence their child's dietary behavior. | This conceptual model describes parental influences on child dietary behavior. Parents influence child home food environment by both their own dietary behavior and food parenting practices. Parent context (parenting styles, differential parental treatment) also mediates this relationship. The home food environment is directly related to child dietary behavior, which is mediated by child characteristics (temperament and appetitive traits). | Active/ Passive | Larsen et al., 2015 [82]. |
| 72 | Fig. 1. Conceptual framework of automatic and underlying techniques that may bridge the intention-behavior gap in food parenting. | This model portrays the pathways of food parenting intentions on food parenting behaviors that influence impulsive food behaviors. Parental habits are influenced by food (cues) and children's emotion/eating and these serve as a moderator for the food parenting intention-behavior gap. | Active/ passive | Larsen et al., 2018, [83]. |
| 73 | Figure 1. Integrative model for understanding acculturation and Latino adolescent mental health. | This theoretical model integrates the multiple influences that environmental, individual, and family factors have on acculturation and Latino adolescent mental health. | Active | Lawton & Gerdes, 2014 [84]. |
| 74 | Figure 2 Influences on PA and sedentary behaviors of preschool-age children organized within the social ecological model. Adapted from McLeroy et al. (1988) | This model represents the conceptual framework on which this review is based (Figure 2; McLeroy, Bibeau, Steckler, & Glanz, 1988) It examines influences on preschool-age children's PA by level. Levels of influence include intrapersonal, interpersonal, environmental, organizational and policy. These levels can interact with and influence health behaviors. | Active/ passive | Lindsay et al., 2017 [85]. |
| 75 | Figure 1. The theoretical frame that will be modeled by using SEM. | This framework shows the relationship between child, parent, and family characteristics and how these determine parenting behaviors and children's psychological adjustment. Individual/ family characteristics have potential direct effects on child psychological adjustment. | Active/ Passive | Liu, 2003 [86]. |
| 76 | Figure 1. Model for Spanish Adolescents. Figure 2. Model for Immigrant Adolescents. | This model demonstrates the positive relation between perceived family support by Spanish and immigrant adolescents to their psychological adjustment, which in turn is positively related to school adjustment. This relates negatively to problem behaviors. Adolescents' psychological adjustment can describe the relationship between family support and school adjustment with family support indirectly impacting school adjustment. | Active/ passive | Lopez-Rodriguez et al., 2018 [87]. |

(*Continued*)

**Table 1.** (Continued)

| # | Model name | Model description | Child's Role | Source (See references for full citation) |
|---|---|---|---|---|
| 77 | Figure 1. Theoretical model linking co-parenting and parent and child anxiety. | This model outlines potential mechanisms of parent and child anxiety with co-parenting and parenting. It demonstrates that parental anxiety may directly interfere with positive co-parenting; it may be related to general relationship problems and also result in elevated concerns about child activities and exposure. With its specific focus on anxiety, the model provides a useful description of the role of co-parenting in child emotional security and anxiety. | Passive | Majdandžić, et al., 2012 [88]. |
| 78 | Figure 1. Conceptual model of the association between fathers' involvement and individual psychosocial health outcomes mediated by family flexibility and moderated by marital quality. | This model seeks to understand the associations between father involvement and the psychosocial health of individual family members including fathers, mothers, and their adolescent children in military families. This model also illustrates how relational characteristics including family flexibility and marital quality can have an impact on these associations. | Active | Mallette et al., 2020 [89]. |
| 79 | Figure 1. Coding schema derived from original model of adolescent asthma self-management | This model explains the intrapersonal and interpersonal factors that influence self-management behaviors that impact asthma outcomes in order to understand how teens managed their asthma. It reflects teen, parent, clinical and researcher perspectives of asthma self-management. | Active | Mammen et al., 2018 [90]. |
| 80 | Figure 4. Revised model of self-management, with delineation of subcomponents specific to asthma and adolescents. | This model displays a revised conceptual model of asthma self-management, outlining the key constructs of self-management processes (assess—decide—respond), tasks (monitoring, managing, communicating, preventing), intrapersonal factors, interpersonal factors, and outcomes. | Active | Mammen et al., 2018 [90]. |
| 81 | Figure 1. The six metathemes of family preparedness, based upon thematic analysis of interviews with families and clinicians. | This model illustrates how six components of preparedness influence cognitive and emotional preparedness for care transitions in the PICU. These components include content of preparatory information, care coordination and delivery of preparatory information, course of care, family, background, coping skills, and support systems, emotional context and care environment. Quality in one component of preparedness can help families feel prepared even if they do not feel prepared in any of the other components. | Passive | Markwalter et al., 2019 [91]. |
| 82 | Figure 1. Partial indirect effects model. | This model quantifies potential mediation of the relationship between adolescent ADHD symptoms and depressive symptoms by maternal and paternal support. Parent support variables examined in the model include involvement, autonomy, and warmth. | Passive | Meinzer et al., 2015 [92]. |
| 83 | Fig. 1. Conceptual model of hypothesized benefits of a bedtime routine. | This model seeks to explain how bedtime routines can positively influence development language development, literacy, child emotional and behavioral regulation, parent-child attachment, and family functioning, mood/emotional/behavioral regulation and sleep. In this model child factors, family factors and other contextual factors influence bedtime routines. | Active | Mindell & Williamson, 2018 [93]. |
| 84 | Figure 1. Theoretical model of children's developing health lifestyles. | This model illustrates the family factors that influence pathways that lead to early adolescent health lifestyle. Family factors like background, resources and parenting influence health lifestyle at school entry, school factors and peer health lifestyles. | Active | Molborn & Lawrence, 2018 [94]. |
| 85 | Figure 1. Phenomena and categories-Campinas, 2016/2017. | This model represents the feelings that grandparents of grandchildren in the PICU experience. Grandparents often report experiencing fear of their grandchild's death, feelings of uncertainty, isolation and suffering. At the same time, they report fighting to anchor the family, provide support and strength, and offer hope for better days. | Passive | Moraes & Mendes-Castillo, 2018 [95]. |

(*Continued*)

**Table 1.** (Continued)

| # | Model name | Model description | Child's Role | Source (See references for full citation) |
|---|---|---|---|---|
| 86 | Figure 1. Levels of interacting family environmental subsystems (LIFES). | This model details the interaction of different types and levels/systems and sub-systems that family environmental influences have on children and adolescent's energy balance-related behaviors (EBRB). This model seeks to explain the various family environmental influences that are interrelated using ecological and systems theories. | Active | Niermann et al., 2018 [96]. |
| 87 | Figure A7.3. Theme material conditions with categories and concepts. | This model displays the material conditions related to work, affordability, and foodscape that impact childhood obesity in England and are related to social class and possibly food-related obesity policy between state and social class. | Passive | Noonan-Gunning, 2018 [97]. |
| 88 | Figure 1. Pathways by which maternal employment may play a role in maternal and child weight status. | This model illustrates the pathways that maternal employment status can lead to changes in maternal and child weight status and BMI. Maternal employment may result in changes to food purchasing, improved household well-being, changes in mothers' time allocation, and psychological effects leading to changes in health and weight among women and children. | Passive | Oddo et al., 2018 [98]. |
| 89 | FIGURE 1 \| A framework for researching the outcomes of family separation due to paternal deportation. | Using an eco-cultural framework, this model offers a conceptual framework to examine potential impacts that paternal deportation has on families' left behind in the U.S. This model considers how pre-deportation family/household context and migrant characteristics influence post-deportation family/household context, which influences post-deportation family/household outcomes and ultimately post-deportation migrant outcomes. It also considers how federal/state policy environment, local climate and immigration enforcement environment influence post-deportation family/household outcomes and post-deportation migrant outcomes. | Passive | Ojeda et al., 2020 [99]. |
| 90 | Figure 3.1. Initial conceptual model. | This model shows a process that links socioeconomic background to family process. Socioeconomic characteristics shape family structure, activity, and social networks, which consequently determine the physical, emotional, behavioral, and economic environments in which a child lives. These environments influence child health. The model shows how distal factors affect child health through more proximal factors experienced directly by the child. | Passive | Panico, 2012 [100]. |
| 91 | Fig. 1. Conceptual model. | This model proposes that family structure trajectories impact child health. It posits that socio-economic pre-cursors such as physical environment, emotional environment, health behaviors and economic environment can affect child health depending on how they are experienced by the child, and are mechanisms that help to explain the relationship between family structure and child health. | Active/passive | Panico et al., 2019 [101]. |
| 92 | Figure 1. Conceptual model for predicting the health-promoting behaviors of children from low-income families. | This model represents three ecological levels and the variables at either a group level or individual level that can influence health promoting behaviors in children from low-income families in South Korea. Ecological levels included intrapersonal (characteristics of low income children), interpersonal (family and peers) and institutional factors (community child care centers). | Active/Passive | Park, 2018 [102]. |

(*Continued*)

**Table 1.** (Continued)

| # | Model name | Model description | Child's Role | Source (See references for full citation) |
|---|---|---|---|---|
| 94 | Fig. 1. Conceptual model to explain motivations of maternal handwashing behavior in the neonatal period. | This conceptual model offers an explanation for the determinants of maternal handwashing behavior in the neonatal period. It is based on the Health Belief Model, and includes perceived advantages and disadvantages of handwashing, normative beliefs and subjective norms, perceived risk and perceived behavioral control as drivers to nurture intention to improve handwashing behavior. This model is also based on the Theory of Reasoned Action/ Theory of Planned Behavior and includes maternal self-efficacy and handwashing intention as motivators for habit, handwashing behavior, actual control and cue to action. | Passive | Parveen et al., 2018 [103]. |
| 95 | Fig. 1. A conceptual model linking individual parent characteristics, parental coping, and individual child characteristics to the management and outcomes of type 1 diabetes (T1D) in very young children (YC-T1D). | This model illustrates how individual characteristics of parents can influence parental coping with affective, behavioral, and cognitive challenges associated with having young children with Type 1 diabetes. The effectiveness of parents to cope with affective, behavioral and cognitive challenges in their children with Type 1 diabetes has an impact on Type 1 diabetes management behaviors and individual child characteristics. | Active/ passive | Pierce et al., 2017 [104]. |
| 96 | Figure 1. Risky families model. | This model depicts the behavioral and biological consequences of risky family environments. Risky family characteristics start a process in early child life that creates vulnerabilities that render children susceptible to adverse events and mental/ physical health problems later in life. It models how family social context and genetic factors interact to influence family social environment, which consequently impacts child stress response, emotion processing, and risky health behaviors throughout development. All of these play a role in mental and physical health problems, which usually manifest in adolescence | Passive | Repetti, Taylor, & Seeman, 2002 [105]. |
| 97 | Figure 1. Conceptual model for reducing health-risk behaviors in middle childhood. | This is a conceptual model for reducing health-risk behaviors in children. It shows that there are risk factors for child health behavior on the family, individual, and environmental level. Both nonmodifiable (i.e., demographic traits) and modifiable (i.e., child traits, parenting behaviors) family and child factors determine health risk behaviors of children. The relationship between risk factors and health outcomes is hypothesizes to be mediated by parent- child communication processes, which can serve to either promote or discourage child health risk behavior participation. | Active/ Passive | Riesch, Anderson, & Krueger, 2006 [106]. |
| 98 | Figure 1. Conceptual model SEM; Figure 2. Results for the model with satisfaction with migration; Figure. 3 Results for the model with desired migration. | This model explores the role of family dynamics on the relationship between migration, economic pressure, and child functioning/ life satisfaction. The influence of migration and pressure on parental and social support, family conflict, and parenting behaviors is examined, and the impact of these variables on child psychological functioning, educational achievement, and satisfaction with life is quantitated. | Active/ Passive | Robila, 2011 [107]. |
| 99 | Figure 1. Theoretical stress process model with family cohesion and family reframing coping as mediators of the influence of family drinking problems and multiple family risks on child mental health with hypothesized direction of relationship arrows; Figure 2. SEM. | This model illustrates the directional associations between child mental health and a number of family variables. It examines how family drinking problems, multiple risk, and negative life events often determine negative mental health symptoms in children, but how family cohesion and coping can mitigate the effect of harmful exposures. | Passive | Roosa, Dumka, & Tein, 1996 [108]. |

(*Continued*)

**Table 1.** (Continued)

| # | Model name | Model description | Child's Role | Source (See references for full citation) |
|---|---|---|---|---|
| 100 | Figure 2. An illustration of how ontological and epistemological choices lead to different routes in universal parenting training. | This model illustrates the difference between preventing and promotive approaches to universal parenting training. Interventions that use a top down approach and focus on risk prevention parent training, including parental views and capacities for child health and well-being. Alternatively, interventions that use a bottom up approach focus on health promotion parent training and include children's own experiences and knowledge and rights of the child. | Active | Rooth, 2018 [109]. |
| 101 | Fig. 2. Path analysis model of the moderating effect of future orientation (family) on the association between bereavement and externalizing problems. Fig. 3 and 4. Path analysis model of the moderating effect of parent-child relationship (Fig. 3) and parental monitoring (Fig. 4) on the association between bereavement and externalizing problems. | This model seeks to explain the moderating effects of future orientation at the individual and family level, parent-child relationship, and parental monitoring on the association between bereavement and externalizing problems. The model represents an ecological/transactional framework and illustrates the impact that protective factors for bereavement have on problem behaviors in adolescence. | Active | Sasser et al., 2019 [110]. |
| 102 | Figure 1. Hypothesized research model Figure 2. Model with standardized Beta values[ab]. Figure 3. Model with statistically significant pathways[ab]. | This model hypothesizes that healthy eating behaviors are influenced by effective parent-child communication in childhood. Parental attitude, subjective norms and perceived behavioral control have will have an impact on parent-child communication and ultimately eating behaviors in emerging adulthood. | Active | Scheinfeld & Shim, 2017 [111]. |
| 103 | Figure 2–1. Social context of child health. | This is a comprehensive model of how the family in context shapes child outcomes. Family functioning is determined by family characteristics (sociodemographic, structure, individual members) in addition to adaptation to external environmental forces (family social network, community, and social policy) and family lifecycles. Family functioning, in turn, influences a child's innate biological and psychological characteristics in parallel with the child's community and development. Influences on the family system are complex; family and child outcomes are multi-faceted. | Passive | Schor & Menaghan, 1995 [112]. |
| 104 | Fig. 1. An integrated conceptual framework of HPV vaccination. | This model illustrates modifying factors, individual health beliefs and cues to action that influence parents' HPV decision making based on six stages in the Precaution Adoption Process Model. Parents' HPV decision making moves from stage 1, unaware, to stage 2, unengaged (vaccine hesitancy), stage 3 undecided, stage 4 decided not to act or stage 5 (anceptor), decided to act and then stage 6 (anceptor), acted. | Passive | Shapiro et al., 2018 [113]. |
| 105 | Figure 1. Aspects of family life. | This model proposes aspects of family life that an effective intervention should target in order to achieve optimal child functioning, with a focus on preventing aggressive and violent behaviors among youth. Family, parenting, and family relationships with other social contexts can all impact child functioning outcomes; these relationships can be influenced by moderating events such as life stress. | Active/ Passive | Smith et al., 2004 [114]. |
| 106 | Figure 2.1. Social context of family health. | This model demonstrates the many social and environmental aspects that contribute or influence family health. It includes external factors such as social policy, physical environment, employment, and the ways that social supports influence family characteristics and internal home processes. Family patterns of interaction directly impact family health promotion and consequently, family health. There is a dynamic interplay between external socio-contextual factors and the inter-home and family environment. | Passive | Soubhi & Potvin, 2000 [115]. |

(*Continued*)

**Table 1.** (Continued)

| # | Model name | Model description | Child's Role | Source (See references for full citation) |
|---|---|---|---|---|
| 107 | Figure 1. Conceptual model of child television exposure. Figure 2. Model estimates. | This model seeks to describe how neighborhood quality, parent social support, and parent stress are potentially important social and familial factors for understanding child TV exposure. Neighborhood quality, social support, and parent stress can influence child TV exposure. | Passive | Swindle et al., 2018 [116]. |
| 108 | Figure 1. Conceptual model of the mediation of the association of kinship support and adolescent well-being. [Figures 2, 3 and 4 are SEM testing of model components.] | This model illustrates how maternal well-being and parenting practices are positively enhanced by kin social support. Better adjusted mothers and positive parenting behaviors can result in more adequate functioning in adolescents. | Active/ Passive | Taylor & Roberts, 1995 [117]. |
| 109 | Fig 1. Conceptual framework of the effects of childhood psoriasis on parents of affected children. | This model represents the negative impact that having a child with psoriasis has on parents emotional well-being, health and self-care, family and social function, and personal well-being and life pursuits. | Passive | Tollefson et al., 2017 [118]. |
| 110 | Figure 1. A conceptual model for family food systems in households with adolescent female athletes. | This model provides a framework for understanding and identifying factors involved in the formation of family food routines and eating activities. Food choices and mealtime routines are directly influenced by the needs (i.e., need for a high carb/ protein diet for athletes) and values (value eating food that will keep family healthy) of family members. Other family characteristics, such as demographic traits and social support also influence food choices and mealtime behaviors. | Active/ Passive | Travis, Bisogni, & Ranzenhofer, L, 2010 [119]. |
| 111 | Fig 1. Questionnaire variables within the International Classification of Functioning, Disability and Health (ICF) framework (ACQoL–Adult Carer Quality of Life; HBSC–Health Behavior of School-age Children Study; MPSS–Measure of Perceived Social Support; MQ–Measurement Question; MSK–musculoskeletal; NIV–non-invasive ventilation; NMD–Neuromuscular Disorder; PEG–Percutaneous endoscopic gastrostomy, QYPP–Questionnaire of Young People's Participation; WEMWBS–Warwick Edinburgh Mental Wellbeing Scale.) | This model seeks to understand non-ambulant youths' mental wellbeing and relationships with physical health, participation and social factors. The model represents a compilation of validated scales and de novo questions. | Active | Travlos et al., 2019 [120]. |
| 112 | Figure 1. Standardized parameter estimates for proposed theoretical model: "Model 1." | This conceptual model links parental physical activity orientations (engagement in activity, attitudes, and perceived importance) to child physical activity and their perceptions of self-efficacy. | Active | Trost et al., 2003 [121]. |
| 113 | Fig. 1. Conceptual model of parent physical activity and screen media practices and beliefs. | This model demonstrates how parental attributes and their perceptions of child physical activity attributes influence physical activity practices in both parent and child and also influence screen media practices. Physical activity practices and screen media practices are influenced by parental permissiveness/neglect, structure, and autonomy support/ responsiveness for their child to engage in these practices and this determines child physical activity and screen media behaviors. | Active/ Passive | Vaughn et al., 2019 [122]. |
| 114 | Figure 1. Proposed model of influences on parent mental health, parenting practices and children of parents with intellectual disability; Figure 2. Final model of influences on parenting and on children of parents with intellectual disability SEM. | Figure 1 presents the influences on parental mental health and subsequent effects on child well-being. Socioeconomic position and social supports (partner support, access to other support) influence parent mental health. Parent mental health is suspected to affect child well-being both indirectly (through parenting style) and directly. Figure 2 explores associations between socioeconomic disadvantage, social support, parent mental health, parenting practices, and child well-being in families where parents have an intellectual disability | Passive | Wade, Llewellyn, & Matthews, 2015 [123]. |

*(Continued)*

**Table 1.** (Continued)

| # | Model name | Model description | Child's Role | Source (See references for full citation) |
|---|---|---|---|---|
| 115 | Figure 1. Mediation and moderation model of the relationship between family structure and child outcomes. | This model illustrates how several family level variables interact to influence child outcomes. The focus is on potential mediators of the relationship between family structure and child outcomes, including social support, context, parent functioning, family relations, etc. Family structure is not expected to have a direct impact on child outcomes. | Active/ Passive | Wise, 2004 [124]. |
| 116 | Figure 20.2. The biobehavioral family model. | This model seeks to examine pathways by which family relations influence disease in children. Family functioning, interactions between spouses, child-parent dyads and external individuals shape emotional climate of family and child life. This family environment determines in part parent child attachment, which ultimately influences child disease state through biobehavioral reactivity (psychological process resulting in biological response to stimulus). | Passive | Wood & Miller, 2005 [125]. |
| 117 | Figure 8.2. Detailed conceptual model based on Nigerian data. | This model describes how family resources indirectly influence child growth and development through family management, beliefs, and caring behaviors. Different components of family life and behavior (material resources, health practices, emotional climate, academic stimulation) are describes and linked to child growth, IQ, SQ [social], mental, and overall outcomes. | Passive | Zeitlin et al., 1995 [126]. |
| 118 | Fig 1. Hypothesized path diagram of parent-child OHS model<br>Fig 2. Path diagram of family OHB model.<br>Fig 3. Path diagram of family OHS model.<br>Fig 4. Path diagram of mother-child OHS model. | This conceptual model reflects the pathways from socioeconomic status, oral health knowledge, attitudes, and practices to the oral health standards of parents and parental influences on children's oral health practices. It is tested using a structural equation model (SEM). Pathways include SES, oral health knowledge and attitude, parents' oral health behaviors and oral health status, parents oral health knowledge and attitude towards children, children's oral health behaviors and children's oral health status. It asserts that parents have a strong influence on their children's oral health. | Active/ Passive | Zhang et al., 2020 [127]. |
| 119 | Figure 2. A conceptual model depicting the relationship between antecedents, attributes, and positive consequences of illness acceptance in adolescents | This is a comprehensive model of illness acceptance in adolescents, which illustrates the relationship between antecedents, attributes, and positive consequences. Antecedents include parental acceptance, peer and family support, developmental readiness, disease education. Attributes include understanding of illness, overcoming limitations, normalization and readiness for responsibility. Positive consequences include higher self-esteem, stronger sense of identity, better disease control, and improved quality of life. | Active | Zheng et al., 2019 [128]. |

health experiences and trajectories, which is described in more detail in Table 6; and (4) a reference for each model. Please note that many of the authors displayed their models in different ways in order to highlight different analyses. As long as the overarching model in any given paper was the same, it was counted only one time even though it may be have been reflected by several distinct figures.

## Description of studies by discipline

Of the 118 unique models identified, 11 broad disciplines were represented in terms of the area of study. This broad range of disciplines, described in Table 2, illustrates the breadth of interest in understanding the multi-dimensional factors that shape family health in a wide range of contexts.

**Table 2. Description of studies by discipline.**

| Main discipline* | Subcategories (including number of studies in each subcategory) | # of studies |
|---|---|---|
| Biosocial sciences | None specified | 4 |
| Family Studies | Early childhood studies; family sciences | 26 |
| Health Studies | Health promotion (6); behavioral and community health (2); health communication (1); health behavior (2); determinants of health (4); population health (3); public health (8); health sciences (2); health education (1); health systems (1); family and/or child health (8); occupational health (2); paediatric health (4); population science (2); human development (3); epidemiology (2); nutrition (food and nutritional sciences) (9); obesity (3); physical activity/human movement (5); disease control (6); society and health (1); health literacy (1); prevention science (2); bereavement (1) | 74 |
| Oral health | Oral health (2); paediatric/preventative dentistry (2). | 7 |
| Nursing | Family health nursing (1); biobehavioral nursing (1); nursing research (4); nursing biobehavioral sciences (1); psychiatric nursing (1); family health care nursing (1); general nursing (4); pediatric nursing (1) | 14 |
| Psychology | (Family) psychology (8); clinical psychology (1); sociology (3); social sciences (1); family psychology (6); community psychology (1); clinical child and family psychology (2); psychology (3) | 25 |
| Education | None specified | 2 |
| Policy | Policy/marriage policy (2) | 3 |
| Disability research | None specified | 2 |
| Child development | None specified | 5 |
| Medical sciences | Medicine (4); oncology (1); primary care research (1) | 8 |

*N.B. Many of the studies were cross listed by more than one discipline, which explains why the number of studies is higher than the 113 manuscripts/chapters that were identified in our search.

## Family characteristics and behaviors identified in models

The family characteristics and behaviors that were identified in the models collectively are described in Table 3.

## Environmental and/or ecological factors described in models

The environmental and/or ecological factors that were described in the models varied. Some focused more on social and physical health determinants and others emphasized intrapersonal and interpersonal health determinants. In all models, multiple levels of influences were described as having an impact on family health, health behaviors and health outcomes. Table 4 displays these ecological factors. To see how these environmental and ecological factors map onto each individual model, please see S2 Table (S2 Table Ecological factors and models).

## Core characteristics of health promoting families

Table 5 presents core characteristics of health promoting families as observed through our next analysis. While the models prioritized positive characteristics, many of the models also offered what we have described as characteristics of "health threatening families." These health threatening characteristics were sometimes directly yet conversely related to the health promoting characteristics. Illustratively, family stability and positive mother and father relationship were identified as health promoting characteristics while interparental conflict and having an unsupportive family were health threatening characteristics. While each family is

**Table 3. Family characteristics and behaviors identified in the models.**

| Characteristic or behavior | Further Description |
|---|---|
| Maternal characteristics | Education, age, marital status; genetics; mental health; obstetric health and birth outcomes; engages with child; prenatal care and nursing; mother's ideologies about women's roles; economic independence; maternal affection; SES |
| Paternal characteristics | Present in child's life/involvement; behaviors and characteristics; paternal affection; employment status; SES |
| Child's characteristics | Emotional security, adjustment; self-efficacy; mastery; "health promoting actor"; regulation of behavior; recipient of outcomes/parental influence; competence and resilience |
| Family characteristics | Shared values; healthy communication and supportive relationships; attitudes around flexibility, self-efficacy, sense of identity and illness; family routines; family composition and structure; family emotional climate |
| Parents | BMI; self-efficacy; family support; knowledge about nutrition and health behaviors; parenting styles; parent sex; education; positive parenting behaviors; biological parents; parental acculturation; mental health |
| Family composition | Single parent; parents divorced; step-family; no parent; female headed household; teen parent; size of family; extended family involvement; number of siblings; grandparent involvement |
| Family characteristics | Emotional stability; quality of parenting; social support; family communication; family coping; shared values/rituals/culture; boundaries/rules; violence; sibling adaption, coping; unity; flexibility; commitment; communication; spiritual well-being; warmth; distribution of resources; family cohesion; organization; functioning; conversations; conformity; norms and values; family identity and commitment; conflict; family coping; family satisfaction; security; family transitions; family management |
| Potential stressors | Child with a disability; divorce; unpredictable events; absent parent; work-family-child care chaos; teen parent; ill parent; disease; conflict and aggression; abuse; neglect |
| Resources for health | Developmental opportunities; dental insurance |

unique, broad characteristics were universally important. These include holding shared values, having healthy intra-family relationships and communication, and encouraging healthy behaviours. Note that there was no consensus between models on what these healthy

**Table 4. Examples of environmental and/or ecological factors described in the models.**

| Ecological Factors | Examples |
|---|---|
| Biological and psychological factors | Age; size at birth; emotional stability; self-esteem; sex; diet; physical activity; genetics; development; self-esteem; self-efficacy; coping mechanisms; cognitive dimensions; dignity; emotional insecurity; gender; disability status |
| Social, cultural and economic factors | SES; education; marital status; employment; ethnicity; household characteristics; social support networks; social interaction; conformity to rules; health beliefs; family relations; family identity; leisure activities; culture; parental development; family health practices; family variables (family obligations, support, well-being); interparental insecurity; family health risks; active play opportunities; abuse; disease; family meals; ethnicity; women's economic independence; maintenance of culture and traditions; language; bereavement processes |
| Health related factors | Medical and health services; healthcare quality; access to health services |
| Community factors | Neighbourhood quality and safety; school zone; healthy community development; community capacity for partnerships; community support; public transport; community programs; religious involvement |
| Physical Environment | Rural or urban; household characteristics and infrastructure (i.e. toilet, water facilities); healthy physical environment; physical activity opportunities; ecological environment and environmental exposure |
| Policy | Health communication; school food policy; school break practices and policy; social policy; family policy |

**Table 5. Core characteristics of health promoting families.**

| Health promoting family characteristics | |
| --- | --- |
| Health promoting familial values | Shared meaning, history and culture; family rituals; family spirituality; commitment to family unity; ethical values; sense of meaning and purpose, religiosity. |
| Health promoting relationships | Positive mother and father relationship (marriage quality); kin support; maternal care and support; mutual support throughout family; family stability and cohesion; positive parent-child communication; affection and attention; sense of family togetherness and congruence; family bonding; emotional bonding and support; family climate (warmth, respect, love, honesty, trust); family balance and harmony, family relationships with neighborhood, peers and school; relationship skills |
| Health promoting attitudes | Family flexibility (adaptability and compromise, acceptance of difference of personality and opinion); self-efficacy (family and child, child self-perceived competence); autonomy granting; encouragement of child personal development, sense of identity and sense of meaning; non-blaming attitudes; positivity; respect for privacy for family members; positive attitudes about food/diet and parental perceived child weight status and health related feeding goals; parental beliefs about child's participation in physical activity; illness acceptance; encourages hope; parental sense of control; congruity between ideologies and roles in mothers; maternal self-esteem; appreciation and affection; compassionate; sense of humour, |
| Health promoting behaviours and habits | *Food related*: Positive nutritional habits/diet and mealtime habits, breakfast consumption; enhancing parental knowledge about nutrition; purchasing healthy foods, reading food labels, companionship at mealtimes; parental dietary behaviour; preparing health, balanced foods and meals; eating slowly, appropriate serving sizes; portion control; parental feeding style; reduce stimulus for overeating; parent involvement in weight gain prevention; fruit/vegetable intake; breastfeeding. *Activity related*: Regular exercise; physical activity participation (family and child); parental support for child physical activity. *Parenting behaviours*. Positive parenting behaviours; family routine; self and family care; parent role modelling healthy lifestyle and social interactions; healthy parenting (emotional and physical); positive parenting skills; positive communication; family conversation; children having responsibilities including chores; partner support; family caring behaviours; communication frequency/content/quality; boundary maintenance; support enhancement of social skills; parental problem-focused coping; positive family problem solving. *Other*. Leisure time activities; enough sleep; healthy energy balance; safety and precautions; hygiene practices; treatment adherence. |
| Other health promoting factors | *Family*. Family structure; single-family household; family size; emotional health in parent and child; child cognitive and communicative function; number of siblings; presence of caregiver, family resources, extrafamilial support, grandparents' socio-economic background, child satisfaction, community/society/social network, family life cycle, child development, child's innate characteristics, home contextual processes, family emotional climate, academic stimulation, parent-child attachment, parents health status, parents oral health knowledge and attitude toward children, child and parent oral health status; decreased parental and family stress; sibling well-being and adjustment; individual/career/marital/parental development. *Physical/structural environment*. wealth; family resources; access to water and good hygiene; safe physical environment; employment (family and specifically maternal); child's home food environment. *Healthcare related*. Access to health care providers and system; relationship with health care providers and system; health care utilization; health insurance. *Education*. Maternal education; parental education; school achievement; exposure to opportunities; knowledge of disease. *Biological*. Good physical health of child; health status of parents and family members; biological and genetic endowment; parent's BMI; child BMI and waist circumference; obstetric health and birth outcomes. *Behavioural*. Child behavioural and cognitive development; child development; child cognitive and adaptive skills; child self-regulatory processes. |
| **Health threatening family characteristics** | |
| Health threatening familial values | None named. |

*(Continued)*

**Table 5.** (Continued)

| | |
|---|---|
| Health threatening relationships | Interparental conflict; maternal care and support; poor member relationships (between family, parent-child and siblings); family climate (unsupportive, cold); lack of parent-child communication; parent-child insecurity; interparental insecurity; low level of parental time |
| Health-related attitudes | Lack of belief in child and parent competence; lack of healthy dietary beliefs; lack of parental sense of control; incongruity between ideologies and roles in mothers; negativity. |
| Health threatening behaviours and habits | Nutritional habits/improper diet, lack of physical activity, child sedentary behaviors, substance abuse, lack of sleep, parental perceived child weight status, sugar-sweetened beverages, meal patterns, parenting practices, family knowledge, parental food and physical activity behaviours, child behavioural problems, family care, school achievement, family conflict, mother-child interactions, child screen time, child feeding practices, negative feeding practices, parenting negativity, parenting overcontrol, poor communication, adolescent risky health behaviours, parenting practices, family drinking problems, neglectful; dysfunctional parenting. |
| Other health threatening factors | *Family.* Living arrangements; poor emotional health in parent or child; divorce, child with disability; parental depression; increased stressors; child adjustment (defensive, stress response, behaviour); child emotional insecurity in family; parent partner violence; parental depression; difficult child temperament; family constraints; family size; number of individuals living in household; lack of family routines; maternal distress; unpredictable family related events (moving, job loss, substance abuse); domestic violence; multiple partner fertility; parental conflict; household chaos; work-family-child care chaos; stress; lack of emotional support; inadequate and/or inconsistent child supervision; absent father/husband; isolation; increased stressors; single parenting; teen parenting; conflict; neglectful home; lack of opportunities. *Physical/structural environment.* Low SES; low family resources for housing; unemployment; work demands; economic stress; food insecurity; grandparents' socio-economic background; nonmodifiable demographics of family and child. *Healthcare related.* Lack of access to health care providers and system; poor or limited relationship with health care providers and system; lack of prevention and treatment of illness. *Education.* Lack of maternal education; low level of school facilities and environment; parental low education level. *Biological.* Childhood obesity; disease; illness; child BMI and waist circumference; reduced child health; biological life event (i.e. timing of menarche); poor physical health of child; childhood dental caries; family genetic health risk. *Behavioral.* Child behavioural and cognitive development; adolescent ADHD symptoms. *Social.* Few friends; risky family social environment; disruptions to community social networks. |

behaviours would be, and models all had specific foci around behaviours (e.g., dietary behaviours and exercise). Even more consistent across models, regardless of the behavioural focus of the model, was access to basic determinants of health such as socio economic background (and related determinants such as access to nutritious food) and education and positive relationships and support within the family.

## The child's role in the health promoting family

There were variations in the models as to how the role of the child was represented. Thirty-two of the models specifically ascribe a role to the child that positions them as active agents in shaping their own health experiences. Another twenty-nine models represent the child as an individual member of the family but with the child having a less prominent or active role in shaping their own health. We describe this as having an active/passive role. Nearly half of the models (58) depict the child as a passive recipient of the actions of others, and the ecological determinants that surround him or her, and as part of a wider system but not necessarily as an active agent in his and or her own right. Table 6 presents the various ways that the different models present the role of the child in the family. (The specific ways that the child's role is depicted in each model is also noted briefly in column 3 in Table 1).

**Table 6. Child's role in shaping health experiences.**

| Child's Role | Description | Examples | # of Models |
|---|---|---|---|
| Active | Child has an active role in health and health behaviors that is specifically represented in the model. | • Child has active role in adjustment and adaptation (i.e. accepting they have diabetes, compliance) [29, 90, 128]. | 32 |
| | | • Child as health promoting actor [11, 53]. | |
| | | • Adolescent acculturation includes choice and participation in cultural practices, values, identification, stress, coping [32, 84]. | |
| Active/ Passive | Child is mentioned in model but has a less prominent role in shaping health and health behaviors, which are instead regulated by adult caregivers. | • Child screen time, child exercise, child BMI and child feeding practices influenced by parental control [21, 27, 44, 66, 73]. | 28 |
| | | • Child's psychological adjustment is a product of child characteristics, family, and parents [86, 107]. | |
| | | • Adolescent self-reliance, problem behavior, psychological well-being impacted by kinship support [87, 117]. | |
| Passive | Child is not prominently displayed in model and does not have an active role in his/her/their own health. | • Child is recipient of supports that influence resilience [19, 24, 62]. | 58 |
| | | • Causal model of determinants that have an impact on child health [20, 46, 103]. | |
| | | • Maternal and paternal determinants influence child's oral hygiene behavior [17, 49]. | |

## Summary of main findings of the studies

Our search for models related to the health promoting family resulted in the consideration of studies from a very broad range of disciplines, methodological approaches, purposes and perspectives. Whether the study was looking at effects of parental depression [20], weight loss and obesity [66, 77]; academic outcomes [107]; mental health outcomes [105]; dieting and nutrition [82]; the participation of children with disabilities [76], child resilience [19], influences on participation in physical activity [36, 60, 61, 122], or parental perceptions regarding health behaviors for their children [63, 111] the importance—but also the complexity—of the task of modeling the potential of the family in the promotion of health or well-being was acknowledged.

Through our analysis, three main themes were apparent. First, and not unexpectedly, ecological or environmental factors are central components to most models or conceptual frameworks [17, 19, 27, 40, 43, 44, 52, 74, 96, 100]. Yet, the factors that were presented and their relative importance varies among the models. Second, most models were attentive to cultural and other diversities. In doing so, it appeared that authors were being intentional about presenting models that were broad enough to make room for a wide range of differences across family types, and for different and ever-expanding social norms and roles pertaining to families and family life. Rather than focus on what a family looks like, many of the models focused on how the family operates together [23, 58, 66, 75, 87, 125]. And finally, our review drew attention to the way that the role of the child is often presented in models of the health promoting family: less as an active agent and contributor to his or her own health within a family and more as a passive recipient of health that is shaped by a complex range of contexts.

## Discussion

### Environmental factors are important but their conceptualization varies by context

A strong similarity among most of the papers and models we reviewed was the priority given to ecological frameworks or approaches when considering the health promoting nature of

families. Overwhelmingly, authors argued that human behaviors and health outcomes cannot be understood without taking into consideration the contexts in which they occur [21, 76, 82, 119]. This kind of thinking was integrated into most of the models, and the ways that each family interacts with various contextual aspects were described as influencing family functioning and health outcomes for all family members. Indeed, individuals within family systems not only influence each other, but are simultaneously influenced by interactions between family members and the environment [21]. Illustratively, the model by Fisher-Owens et al. (2007) depicts community, family and child level influences as important in shaping child oral health [55]. These authors elucidate their model by describing how the influences on oral health do not act in isolation but rather dynamically, via complex interactions. In 2017, Kalil [72] used Fisher-Owen's et al. (2007) model to further posit that these community, family and child level influences are bound by time and environment as complex interactions in which children live and experience their lives, and they have an impact on child oral health. In their 2014 model, De Coster and Zito demonstrate the importance of contextual factors by describing how emotional attachment of young people to their mothers is shaped by maternal distress, which in turn influences adolescent mental health outcomes [41].

The importance of environmental or ecological factors is well-established in the academic literature [1, 2], and our observation about their importance in these models is hardly groundbreaking. What is interesting about our findings, however, is that while there were variables that were seen in models repeatedly (for example, SES, family organization, etc.), there was no real consensus about what the actual environmental factors that were important to the various models might be. In part, identifying environmental factors that are important is complicated by the importance of contextually and culturally appropriate measurement and interpretation; what is a valid measurement or factor in one context may be interpreted differently in another. For instance, the environmental, individual and family factors related to acculturation in adolescent Latino [84] and Spanish [87] immigrant mental health differ from the influences related to youth mental health and parental risk taking, alcohol dependency, or single parent households [108, 123]. The issues that appear to shape the influence of parents over their child's mental health are different in different cultural contexts. While in all of these models [84, 87, 108, 123] child/adolescent mental health is influenced by parental and family variables, in some models, parental variables are predisposed by culture and context. Illustratively, in some contexts, acculturation [84] and immigration [87] are important shaping factors on youth/adolescent mental health, in other contexts these are not relevant. From geographic and cultural contexts as far ranging as rural northwest China [86], Romania [107], Latino youth in the United States [84], South Africa [78], South Korea [102], Kenya [42], Spain [87], African American [57], and Uganda [69] complex and dynamic relationships between various aspects of the child and family environment were characterized in diverse ways. The conceptual frameworks that were developed were influenced by geographic and cultural contexts. One of the challenges of developing a conceptual framework for the health promoting family, and which indeed was recognized strongly in the studies in this review, is the importance of acknowledging that cultures, contexts, and families are unique. So too are at least some of the environmental factors that contribute to family well-being [24, 85, 91, 96].

Despite these natural contextual variations, the environmental and/or ecological factors that were described in the models mapped readily onto already well established social, physical, and structural determinants of health. Overall, while not surprising, our review suggests that researchers continue to find and use determinant of health frameworks when developing conceptual models related to family health [31, 46, 103]. While each family is unique, as our analysis in Table 5 demonstrates, there are other broad characteristics that appear to characterize family health. These include shared values (it does not matter what

the values are so much as that they are shared); positive relationships; attitudes that support positivity, flexibility, care and healthy behaviours; access to basic determinants of health such as sufficient income and other health resources and access to healthcare. Table 5 also includes an analysis of health threatening family characteristics and includes factors such as family and interparental conflict; negative health behaviours (improper diet, lack of sleep and physical activity; family substance problems) and lack of basic determinants of health such as insufficient income; food insecurity and lack of access to health care providers and healthcare relationships. This review was prompted by our observation that a universal definition of a health promoting family does not exist. This scoping review reinforces the complexity of providing such a definition. Yet, what it does contribute is a synthesis of some of the basic categories and characteristics of health promoting (and health threatening) features of families, even in their uniqueness.

## Diversity, and changing norms around social roles

Over the past many decades, dramatic societal shifts have occurred around norms of family life (including, for example, shifts in social and employment roles of men and women [28, 41], and the role and status of women overall). These societal changes include a resistance to restrictive paradigms about what it is to be a family, and a growing recognition that families come in many shapes, sizes and configurations. This makes it difficult to determine what a healthy family might look like in a diversity of contexts, and perhaps more importantly, reveals not only the pointlessness but also the danger of prescribing a typical family life cycle too specifically. This is especially true as families inevitably have expected or unexpected transitions over the life span. The focus we see in this literature review away from what "constitutes" a family to how a family operates is certainly healthy and avoids claims of any false normal.

As thinking around health and families evolve in ways that decentre what may be considered "normal", it draws attention to how understandings of health have evolved. This, too, was reflected in our review. Illustratively, Ball, Moselle & Pedersen (2007), point to the way that as understandings of health have expanded, "scholars and policy makers focused on families are increasingly subscribing to understandings of health as reciprocally determined by a broad array of biological and non-biological factors" [23, p. 6]. Notably, Denham (2003) [43] encourages thinking that moves beyond Western, dualistic and biomedical foci on health, illness and disease to a consideration of more diverse ways to approach individual and family health.

Consideration of adult gender was important across the models. It was then surprising that it was not as big a consideration in relation to the children in the majority of the models. However, where gender was considered, it was important. Illustratively, in their model, Molborn & Lawrence [84] draw attention to the overall weakening of socioeconomic disparities in health lifestyles and a strengthening of gender disparities as children age. Niermann et al. [96] model gender differences in the association between family functioning and weight status. While a higher level of family functioning was associated with decreased likelihood of being overweight among girls, this was not the case for boys. In the 2018 model by Shapiro et al., [113] there was a significant association between child's gender and the Precaution, Adoption, Process Model (PAPM) stage of decision-making, with parents of boys more likely to report being in earlier PAPM stages. Here, parents of daughters (compared to sons), parents of older children, and parents with a health care provider recommendation had decreased odds of being in any earlier PAPM stage as compared to the last PAPM stage (i.e. decided to get vaccinated). None of the models made any room for gender diversity or non-binary gender. We would expect as models of the family continue to evolve, attention to non-binary gender among all family members will become much more prominent in future models.

## The child as a health promoting actor is undervalued

In our analysis of these models, the lack of attention to the kind of robust vision that was cast by Christensen in 2004 [10] as to the value of the child as health-promoting actor in these models was striking. Admittedly, and as depicted in Table 6, 32 (out of a possible 118) of the models that were reviewed did present children as active participants in achieving their own health. For example, both Gold et al. (2008) [59] and Wade et al. (2015) [123] noted self-efficacy as important to their model and Hauser-Cram et al. (2001) [64] drew attention to the child's ability to attain mastery and also to regulate one's own behavior. We were interested to note that gender did not appear to be a consideration in terms of the child's active or passive role. Age, however, appears to be important. In the 32 "active participant", older children and adolescents were more likely to be described as having an active role than younger children. This is not surprising given that as children and youth age, they naturally begin to take a more independent role in their own health. Several studies drew attention to the child's role in avoiding risk behaviors such as risky tobacco and alcohol use [5, 17, 35, 40, 51, 94, 96, 97, 99]. While another 28 of the models presented children's roles in what we categorized as "active/passive" roles, more often, however, these models (58 out of 118) presented children as passive recipients of health rather than as contributing agents to their own health journeys.

This lack of attention is short-sighted, because as Christensen [10] and others [129] [130] have argued, when children themselves are not included and encouraged as competent, capable agents, they are deprived of the opportunity to learn to make their own health related decisions, and to gradually learn to take responsibility for their own health behaviors and decisions. Including the child in this way is not intended to diminish the importance of the role of the parent(s) or environmental and contextual factors in shaping the health trajectories of children. Rather, it is in keeping with a growing body of research that illuminates the importance of children's contributions to the health promoting nature of their own families, and the empowerment that ensues when children are encouraged to contribute to the health promoting activities in the family [80, 129, 130]. In keeping with this scholarship, Woodhead and Faulkner [131] use research evidence to describe how the emergent competencies of children are not so much set along an artificial developmental timeline as they are *grown into* through active participation. When children are guided in their participation by supportive adults, their developmental capabilities evolve. In other words, when children are encouraged to become active agents in their own health journey, their participation itself appears to serve the dual purpose of also supporting their development [131].

One area to which this scoping review draws attention is in relation to illness acceptance, maintenance and self-management behavior in adolescents, and the ways that these kinds of active roles can be of particular importance [90, 128]. For instance, in their model, Mammen et al. (2018) [90] describe how self-management behaviors are motivated by personally important outcomes in teens related to their own ideas about symptom perceptions, medication beliefs, symptom management, and personal goals and priorities. Additionally, Zheng et al. (2019) [128] describe how the active roles that adolescents play in terms of understanding of their illness, overcoming limitations, normalization and readiness for responsibility lead to positive consequences of higher self-esteem, stronger sense of identity, better disease control, and improved quality of life in adolescents. In turn, all of this supports illness acceptance.

We observed a slow but potentially encouraging shift that appears to have occurred over the past five years. Whereas we observed that in earlier models, children were prescribed a primarily passive role (for example, only about ¼ of the models identified before Christensen's model was published in 2004 recognized the child as having an active role), a shift towards recognizing children as active agents in promoting their own health in many of the later studies

was notable. Illustratively, within the 44 models that we identified between 2016 and 2020, over 1/3 of them (16/44) depicted the child as having an active role in promoting family health. It may be that the initial vision Christensen [10] proposed in her original theoretical framework, which includes the child as a health promoting actor, and that was the impetus for this review, is becoming more widely accepted as important to the health promoting potential of family contexts.

The notion of the child as a key health promoting actor in families is in keeping with Article 12 of the Convention on the Rights of the Child (CRC), which outlines participation rights [132]. Children from countries who have ratified the CRC, in keeping with their age and evolving capacities, have the legal right to express their opinions, to have a say in matters affecting their own lives, and to participate fully in society. This enables not only public agency, but also agency in their own family context. Participation as active, health promoting agents in the life of their family is an opportunity by which young people can have their ideas valued and recognized and can influence decision-making in ways that affect their lives. These kinds of roles not only contribute to the life of the family overall, but also facilitate growth, resilience, meaning and agency in the life of the child [71, 93]. This kind of active participation is also an internationally protected right [132]. Consequently, attending to children's voice, agency and participation should remain central to the ways that models of family health are shaped [133, 134].

## Strengths and limitations

To our knowledge this is the first scoping review to identify studies that model the health promoting family. The strengths of this review include the systematic methods used for identifying included models. It provides an overall summary table that demonstrates the diversity of interest in this topic, and the different ways that health promoting families have been modelled across disciplines over decades. A limitation of this review is that only papers written in English were considered and relevant material written in foreign languages were omitted. This inevitably introduced a layer of bias in the final sample of included models.

## Conclusions

In this review, we identified 118 models that describe the health promoting potential of families. The complexity of contemporary family life was well-described, including appropriate attentiveness to rapidly changing social norms and roles. Ecological and environmental factors were given high importance in all models, yet consensus on what the specific factors are that would facilitate a health promoting family rightly remained elusive. The models identified in this literature review come from a diversity of disciplines and indicate a broad and general relevance of family health. This could imply that a broad range of stakeholders are open to considering family health promotion and intervention strategies in a variety of different disciplinary contexts. The role of the child as an active agent—rather than a passive recipient—of their health journey was highlighted as an important gap in many of the identified models. Future research would do well to pay attention to the capacity of children within families to be active agents in shaping their own lives and the lives of their family members [134]. Not only is the active participation of children an internationally protected right, it is a powerful vehicle for supporting the emergent competencies of young people in terms of managing their own health experiences and trajectories.

The family is a key setting for health promotion. Contemporary health promoting family models can be used to establish scaffolds for shaping health behaviors and outcomes for families and can be useful tools for education and health promotion. This review contributes a synthesis of contemporary literature in this area and supports the priority of ecological

frameworks and diversity of family contexts. It also encourages researchers, practitioners and family stakeholders to recognize the value of the child his or herself as an active agent in shaping the health promoting potential of their family context.

## Supporting information

**S1 Table. HPF review evidence table.**
(DOCX)

**S2 Table. Environmental and/or ecological factors detailed in models.**
(DOCX)

**S3 Table. Preferred reporting items for systematic reviews and meta-analyses extension for scoping reviews (PRISMA-ScR) checklist.**
(DOCX)

## Acknowledgments

We would like to thank Chelsea Humphries, participating investigator with the specified role of librarian who consulted on the updated search strategy. We would also like to thank Jessica Byrnes, who was our research assistant throughout much of this project.

## Author Contributions

**Conceptualization:** Valerie Michaelson, Kelly A. Pilato, Colleen M. Davison.

**Data curation:** Valerie Michaelson, Kelly A. Pilato, Colleen M. Davison.

**Formal analysis:** Valerie Michaelson, Kelly A. Pilato, Colleen M. Davison.

**Funding acquisition:** Valerie Michaelson, Colleen M. Davison.

**Investigation:** Valerie Michaelson, Kelly A. Pilato, Colleen M. Davison.

**Methodology:** Valerie Michaelson, Kelly A. Pilato, Colleen M. Davison.

**Project administration:** Valerie Michaelson, Kelly A. Pilato, Colleen M. Davison.

**Resources:** Valerie Michaelson, Kelly A. Pilato, Colleen M. Davison.

**Supervision:** Valerie Michaelson, Colleen M. Davison.

**Validation:** Valerie Michaelson, Kelly A. Pilato, Colleen M. Davison.

**Visualization:** Valerie Michaelson, Kelly A. Pilato, Colleen M. Davison.

**Writing – original draft:** Valerie Michaelson, Kelly A. Pilato, Colleen M. Davison.

**Writing – review & editing:** Valerie Michaelson, Kelly A. Pilato, Colleen M. Davison.

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
