## [Decision Letter · Decision Letter 0]

1 Feb 2021

PONE-D-20-23373

Family as a health promotion setting: 

A scoping review of conceptual models of the health-promoting family

PLOS ONE

Dear Dr. Michaelson,

Thank you for submitting your manuscript to PLOS ONE. After careful consideration, we feel that it has merit but does not fully meet PLOS ONE’s publication criteria as it currently stands. Therefore, we invite you to submit a revised version of the manuscript that addresses the points raised during the review process.

We look forward to receiving your revised manuscript.

Kind regards,

Johnson Chun-Sing Cheung, D.S.W.

Academic Editor

PLOS ONE

2. Please ensure that the time interval for the search is included in the Methods section.

3.In your Data Availability statement, you have not specified where the minimal data set underlying the results described in your manuscript can be found. PLOS defines a study's minimal data set as the underlying data used to reach the conclusions drawn in the manuscript and any additional data required to replicate the reported study findings in their entirety. All PLOS journals require that the minimal data set be made fully available. For more information about our data policy, please see http://journals.plos.org/plosone/s/data-availability.

Reviewers' comments:

Reviewer #1: REVIEW OF: PONE-D-20-23373 – Family as a health promotion setting:

A scoping review of conceptual models of the health-promoting family

General Comments

This detailed scoping review focuses on the role of family as a source of health promotion. It identifies conceptual and theoretical models of family to compare and contrast these models. The stated objective of this systematic review is to provide details on how these models “are being used in health promotion contexts.” Investigators state that their purpose in this scoping review is to identify, “key characteristics related to the concept of the health promoting family.”

The investigators report that their extensive search using PRISMA review criteria which identified 56 manuscripts that meet their review criteria. This process identified 61 unique models. Authors report identifying three main themes: (a) ecological models, (b) models that examine diverse cultural factors, and (c) models that address the role of children within the family as “passive” recipients of “their health journey.” Investigators identify the concept of a “health promoting family” as an important entity in this line of research. In this regard, are there families that consistently engage, in a trait-like manner, in healthful behaviors or is healthfulness a dynamic and changing process?

Specific Comments

Investigators cite the Chistensen (2004) model of families, that refers to the “the health promoting family.” This is regarded as an important construct, although these investigators do not define what constitutes a “health promoting family.” They also do not provide details on the prevalence and core characteristics of such families within various populations. How prevalent are such families within a given environment, and what factors or family system propensities may operate as core characteristics of health promoting families?

Investigators recognize that families are complex and diverse. However, given those wide variations in family systems, based on the empirical data examined in their selected studies, what familial values, health-related attitudes, health behaviors and habits, and other familial factors may be regarded as features that characterize such healthy families? Conversely, health outcomes could instead be regarded as situational, where some families may exhibit propensities towards achieving healthful outcomes, with few families always being healthy, while other families do not always exhibit family pathology and illness outcomes.

The methodology used to identify and select relevant articles for review is conventional for various systematic reviews. This manuscript search was guided by a research question that explores how a family’s “health promoting potential” is “portrayed in emerging conceptual and theoretical models.” Investigators present in Figure 1 their Search String consisting of 22 indicators used to identify relevant articles. Further description is needed to inform readers on how that complex syntax was used in their search process. Many search engines, such as PsycINFO, allow investigators to specify their search terms as they appears within a document’s “key words,” “abstract,” or “anywhere” in the document. Why did these investigators use the noted search strings, rather than the conventional search term procedure?

Page 8. As shown in Table 2, investigators describe their synthesis criteria as this involves stratifying the identified models by authors’ discipline. Table 1 , identifies familial characteristics, and role of the child. These criteria appear to be the same or similar to the emerging themes which these investigators mention. How were these criteria decided, or were the inductively identified as themes? If so, by what process were these themes identified?

Page 9. Table 1 is mentioned which presents 119 entries identified as model figures. Further clarity would be helpful in distinguishing how these 119 entries or models relate to the aforementioned 61 models. Information is summarized regarding: the model name, model description, role of the child, and citation for the source (the first author and year of publication). The model description provides the most detailed information about a given model.

Summarizing Comments

This major multi-year effort is laudatory in its compilation of descriptions for 119 models and the issues examined. A few key findings are mentioned in the Discussion section. These include: (a) the ecological systems emphasis that is integral to most of these models, and is important for understanding family contexts, (b) the recognition of large variations in family models, and (c) the role of the child as a passive or active agent in their role within the family. These investigators note that despite their detailed analyses, “familial health promotion” as generated from these model analyses remains elusive. Nonetheless, the stated purpose of this systematic review was to identify health promotion aspects of these models. Should this purpose be re-stated, or alternately can the author provide more explicit indications derived from their identified models that can attain this purpose?

In their Conclusions section, investigators state that future research should attend to the role of the child as an active agent in own health-related activities. However, the child’s capacity for autonomous self-directed behaviors is related to their age and developmental capabilities. This point is mentioned in the Discussion section. It is notable that these investigators indicate that child or adolescent agency is evident when engaging in certain self-monitoring behaviors. The noted age-related distinctions involving developmental capabilities in childhood, adolescence, and young adulthood needs further elaboration in advocating for a greater emphasis on youths’ role as active agents within the family system.

Investigators comment on the extensive variations in family systems, which makes it difficult to identify core features of family actions that promote health and wellbeing. This is indeed a major challenge. Nonetheless, whereas this challenge is great, the current analyses, while informative, falls short of identifying key factors that can inform the field on major emergent themes that distinguish family function that promotes health, versus that which can be detrimental to health. Further depth of analysis is needed which can draw on this extensive compilation of family models, to identify core features or functions occurring across these family models, that reveal emergent or recurring themes or key factors across models, that are associated with healthful outcomes.

Regarding health outcomes, the construct of “health” in itself is factorially complex. The indication that the concept of health needs to extend beyond a Western dualistic mind-body framing is indeed important. Nonetheless, here as well, this current analyses fall short of making these distinctions in types of health-related outcomes, which also limit these investigators’ ability to relate familial health-related antecedents to specific health outcomes. Instead of classifying the identified models by discipline, which is informative but less useful, these investigators could instead re-classify the identified models in terms of model features, such as risk and protective factors, that are associated with improved health outcomes, as contrasted with health compromising outcomes. Alternately, they might consider another form of model classification, that is more informative for attaining their purpose or objectives as stated for this systematic review.

Reviewer #2: This is an interesting and well written article.

I recommend the authors address the following minor comments:

1- Page 30 line 218: can authors add family characteristics to the model table ?

2- Page 31 line 226: can environmental and/or ecological factors described for the models be connected to the models. For example making signs that show which model/study used these factors.

3- Page 34 lines 255, 259 , and 272 please add citations.

4- Page 35, line 293, what is authors’ proposition?

5- Page 35 line 296, please provide examples.

6- Page 36: lines 300 and 310 the sentence is not clear please re-write and explain better (complex and dynamic relationships between various aspects of the child and family environment guided the conceptual frameworks that were developed).

7- Page 36 line 313 please delete: many

8- Page36, lines 252-254 : did any study evaluate the children age and gender for being active or passive? Where there differences?

9- Page 39 line 372 please delete: did

10- Page 39, discussion for lines 377-379, again did any of those studies factored in child age and gender. If they did could you add to the discussion their findings ?

---

## [Author Response · Author response to Decision Letter 0]

6 Mar 2021

Reviewer #1: 

General Comments: This detailed scoping review focuses on the role of family as a source of health promotion. It identifies conceptual and theoretical models of family to compare and contrast these models. The stated objective of this systematic review is to provide details on how these models “are being used in health promotion contexts.” Investigators state that their purpose in this scoping review is to identify, “key characteristics related to the concept of the health promoting family.”

The investigators report that their extensive search using PRISMA review criteria which identified 56 manuscripts that meet their review criteria. This process identified 61 unique models. Authors report identifying three main themes: (a) ecological models, (b) models that examine diverse cultural factors, and (c) models that address the role of children within the family as “passive” recipients of “their health journey.” Investigators identify the concept of a “health promoting family” as an important entity in this line of research. In this regard, are there families that consistently engage, in a trait-like manner, in healthful behaviors or is healthfulness a dynamic and changing process?

Response: Thank you for this concise summary of our review, and for this interesting question. One of the themes that our review has highlighted is that not only is every family’s health experiences, opportunities, barriers and trajectories unique, each individual within the family (including their individual experiences, opportunities, barriers, relationships) impacts the health of the family overall. Therefore, we would agree with the reviewer’s second suggestion, that healthfulness is “a dynamic and changing process.” While there are health promoting characteristics (including values, behaviors, attitudes, etc) that shape families to be health promoting contexts, many of the factors that we have identified through this review (such as family relationships and roles) change over the life course. What is health promoting at one point in time in the life of the health of one family needs to shift as the needs, values and relationships within the family evolve. This response is woven throughout our manuscript. 

Comment 1: Investigators cite the Chistensen (2004) model of families, that refers to the “the health promoting family.” This is regarded as an important construct, although these investigators do not define what constitutes a “health promoting family.” They also do not provide details on the prevalence and core characteristics of such families within various populations. How prevalent are such families within a given environment, and what factors or family system propensities may operate as core characteristics of health promoting families?

Response: As the reviewer has noted, while we began with Christensen’s model, we do not offer further definition as to what constitutes a “health promoting family” (HPF). One of our main motivations for doing this review was because of the gaps that we observed, including the lack of such a definition. That the reviewer raised this point demonstrates that we were not clear enough in articulating our objectives. We have added to the prose on page 6 (lines 172-174) to make it more clear that we not think that such a definition is available. 

The reviewer’s second point that we have not provided details on core characteristics of such families was also well-taken. After consideration of this comment, and several of the comments that follow, we realized that additional analysis would strengthen this review. This would enable us to make a more meaningful contribution to the literature around what does constitute a health promoting family, even while we still hold that this will vary by context. We have responded by going back to the models and conducing a new analysis, specifically to address some of the gaps that this reviewer has drawn our attention to. This includes analysis of: 1) factors that operate as core characteristics of HPF including familial values, health related attitudes and health beahviours and habits and 2) Health threatening behaviours, including familial values, health related attitudes and health behaviours and habits. We have included this analysis in the main paper as Table 5 (see page 35).

Comment 2: Investigators recognize that families are complex and diverse. However, given those wide variations in family systems, based on the empirical data examined in their selected studies, what familial values, health-related attitudes, health behaviors and habits, and other familial factors may be regarded as features that characterize such healthy families? Conversely, health outcomes could instead be regarded as situational, where some families may exhibit propensities towards achieving healthful outcomes, with few families always being healthy, while other families do not always exhibit family pathology and illness outcomes.

Response: Again, this reviewer is drawing our attention to the need for further analysis so as to provide more information about core charaacteristics of health promoting families. What we learned through this new analysis, which is presented in Table 5 (new to this resubmission), was in keeping with what we had already observed; the most important features that characterize healthy families relate to shared values (it doesn’t matter what the values are so much as that they are shared); positive relationships; attitudes that support positivity, flexibility, care and healthy behaviours; access to basic determinants of health such as sufficient income and other health resources and access to healthcare. This table also includes an analysis of health threatening family characteristics, and includes factors such as family and interparental conflict; negative health behaviours (improper diet, lack of sleep and physical activity; family substance problems) and lack of basic determinants of health such as insufficient income; food insecurity and lack of access to health care providers and healthcare relationships.

We have integrated these new results into our discussion and conclusions.

Comment 3: The methodology used to identify and select relevant articles for review is conventional for various systematic reviews. This manuscript search was guided by a research question that explores how a family’s “health promoting potential” is “portrayed in emerging conceptual and theoretical models.” Investigators present in Figure 1 their Search String consisting of 22 indicators used to identify relevant articles. Further description is needed to inform readers on how that complex syntax was used in their search process. Many search engines, such as PsycINFO, allow investigators to specify their search terms as they appears within a document’s “key words,” “abstract,” or “anywhere” in the document. Why did these investigators use the noted search strings, rather than the conventional search term procedure?

Response: Thank you for your careful review and for this recommendation. We have added a description as an example of the search we conducted in Medline to our methodology section with tracked changes on pages 7-8.

Comment 4. Page 8. As shown in Table 2, investigators describe their synthesis criteria as this involves stratifying the identified models by authors’ discipline. Table 1 , identifies familial characteristics, and role of the child. These criteria appear to be the same or similar to the emerging themes which these investigators mention. How were these criteria decided, or were the inductively identified as themes? If so, by what process were these themes identified?

Response: The reviewer is correct that the criteria we used to synthesize the models are similar to the emerging themes that we present. 

The criteria that we used to synthesize these studies were decided because of their direct relevance to our review questions and objectives. Criteria used in Table 1 stemmed directly from Christensen’s original model, in which Christensen is attentive to familial characteristics and the role of the child. Thus, we grouped the studies by these two categories.

The synthesis in Table 2 included attention to the author’s discipline. This, too, was in keeping with our review objectives in terms of our goal to conduct a broad interdisciplinary survey of previous research. We anticipated that drawing attention to the broad range of disciplines in which the health promoting potential of the family would be valuable to practitioners whose work involves supporting families in a wide range of contexts. The interdisciplinary nature of the work that is being done in this area struck us as an important finding.

The remaining cirteria were developed inductively as we engaged in the analysis and had iterative critical conversations between researchers. We have added text to page 9 (lines 232-242) to make this clear. 

Comment 5. Page 9. Table 1 is mentioned which presents 119 entries identified as model figures. Further clarity would be helpful in distinguishing how these 119 entries or models relate to the aforementioned 61 models. Information is summarized regarding: the model name, model description, role of the child, and citation for the source (the first author and year of publication). The model description provides the most detailed information about a given model.

Response: 61 models is a typographical error. We take full responsibility and are grateful to the reviewer for catching this error. As we did these revisions, we realized that we had counted one model (Park, 2018) twice. Initially we had described 119 models, yet in reality there are only 118 unique models. We have corrected this throughout as well. We have also updated our Scoping Review Flow Chart to reflect this correction.

Comment 6: This major multi-year effort is laudatory in its compilation of descriptions for 119 models and the issues examined. A few key findings are mentioned in the Discussion section. These include: (a) the ecological systems emphasis that is integral to most of these models, and is important for understanding family contexts, (b) the recognition of large variations in family models, and (c) the role of the child as a passive or active agent in their role within the family. These investigators note that despite their detailed analyses, “familial health promotion” as generated from these model analyses remains elusive. Nonetheless, the stated purpose of this systematic review was to identify health promotion aspects of these models. Should this purpose be re-stated, or alternately can the author provide more explicit indications derived from their identified models that can attain this purpose?

Response: Thank you for your kind words. We appreciate this suggestion and have restated our purpose in our discussion section. We have used this opportunity to make our point more explicitly that an important finding from this review was that familial health promotion indeed, is elusive. 

In reponse to this and previous comments from this reviewer, we have also conducted additional analysis of these models. While we still maintain that it would be unwise to prescribe or define a health promoting family too specifically, we have used this analysis (in Table 5) to provide more insights from these models as to the different charactistics that appear to be health promoting and health threatening. 

Comment 7: In their Conclusions section, investigators state that future research should attend to the role of the child as an active agent in own health-related activities. However, the child’s capacity for autonomous self-directed behaviors is related to their age and developmental capabilities. This point is mentioned in the Discussion section. It is notable that these investigators indicate that child or adolescent agency is evident when engaging in certain self-monitoring behaviors. The noted age-related distinctions involving developmental capabilities in childhood, adolescence, and young adulthood needs further elaboration in advocating for a greater emphasis on youths’ role as active agents within the family system.

Response: This is a very thoughtful comment. To address it, we have drawn on scholarship by 

Woodhead and Faulkner (2008). They argue that children’s emerging competencies develop through participation, within supportive relationships. Thus, when young people are provided the opportunity and supports to participate as an active agent in their own health, they are at the same time developing their competencies. We have elaborated on this in the discussion section (pg 46). We also conducted additional analysis of the models in which children were identified as Active Participants. Through this we observed that older children and adolescents were more likely to be described as having an active role than younger children. We discuss this on page 45. 

Comment 8: Investigators comment on the extensive variations in family systems, which makes it difficult to identify core features of family actions that promote health and wellbeing. This is indeed a major challenge. Nonetheless, whereas this challenge is great, the current analyses, while informative, falls short of identifying key factors that can inform the field on major emergent themes that distinguish family function that promotes health, versus that which can be detrimental to health. Further depth of analysis is needed which can draw on this extensive compilation of family models, to identify core features or functions occurring across these family models, that reveal emergent or recurring themes or key factors across models, that are associated with healthful outcomes.

Response: Thank you. We have addressed this comment by adding to the current analyses in order to identify key factors that can inform the field. We do appreciate the challenge and feel that our new table, Table 5 (and related discussion) makes a substantial contribution to this manuscript.

Comment 9: Regarding health outcomes, the construct of “health” in itself is factorially complex. The indication that the concept of health needs to extend beyond a Western dualistic mind-body framing is indeed important. Nonetheless, here as well, this current analyses fall short of making these distinctions in types of health-related outcomes, which also limit these investigators’ ability to relate familial health-related antecedents to specific health outcomes. Instead of classifying the identified models by discipline, which is informative but less useful, these investigators could instead re-classify the identified models in terms of model features, such as risk and protective factors, that are associated with improved health outcomes, as contrasted with health compromising outcomes. Alternately, they might consider another form of model classification, that is more informative for attaining their purpose or objectives as stated for this systematic review.

Response: Again, the reviewer has challenged us to provide more analysis of these models, which we have done and reported in Table 5. Along with other features that we have already noted in this reponse, this new table includes a classification of the models in terms of risk and protective factors for health promoting families. 

We have however retained our table in which we categorize the models by discipline. We think that this table too makes an important contribution as it draws attention to the transdisciplinary interest in this subject. 

Reviewer #2

General Comment: This is an interesting and well written article.

Response: Thank you. We appreciate your encouragement. 

Comment 1: I recommend the authors address the following minor comments: 1- Page 30 line 218: can authors add family characteristics to the model table ?

Response: This is a good suggestion. We have added family characteristics to Table 3 as recommended. 

Comment 2: Page 31 line 226: can environmental and/or ecological factors described for the models be connected to the models. For example making signs that show which model/study used these factors.

Response: This is a good idea, and we took some time to decide how best to demonstrate which model/study used which environmental and ecological factors. Because of the sheer number of models, we did not want to bog down the paper. Yet, we also agree with the reviewer that this is useful information. Our solution was to include this information in a large Supplementary Information (SI Table 2 Ecological factors and models). Here, using the basic model information as a basic frame, we mapped the environmental and ecological factors onto each model. We point readers to this table to view this information. 

Comment 3- Page 34 lines 255, 259 , and 272 please add citations.

Response: We have added the requested citations with tracked changes on page 36 and 37 (now page 39).

Comment 4- Page 35, line 293, what is authors’ proposition?

Response: While there is no consensus across the models as to what the actual environmental factors that are important to families are, our own observation was that even in their diversity, the models mapped consistently onto already well established dosical physical and structural determinants of health. This includes the importance to things like having enough income, health care access, and social supports. We have made this more clear on page 41 of the revised manuscript.

Comment 5- Page 35 line 296, please provide examples.

Response: We have provided examples as requested. These are found on lines 357-367 of the revised manuscript.

Comment 6- Page 36: lines 300 and 310 the sentence is not clear please re-write and explain better (complex and dynamic relationships between various aspects of the child and family environment guided the conceptual frameworks that were developed).

Response: Thank you for this encouragement to make this more clear. We have re-written these sections, as requested. 

Comment 7- Page 36 line 313 please delete: many

Response: This has been done, as requested. 

Comment 8- Page36, lines 252-254 : did any study evaluate the children age and gender for being active or passive? Where there differences?

Response: Gender did not appear to be a consideration in terms of the child’s active or passive role. Age, however, appeared to be important and older children and adolescents were more likely to be descriped as “active” than younger children. We have added text to lines 481 onward to discuss what we found in terms of age and gender. Thank you for asking this question.

Comment 9- Page 39 line 372 please delete: did

Response: We have deleted “did” on this line. 

Comment 10- Page 39, discussion for lines 377-379, again did any of those studies factored in child age and gender. If they did could you add to the discussion their findings ?

Response: We had not noticed any attention to gender in the models in our initial scoping review. After reflecting on this comment, we went back over all of the models again, with a specific focus on analyzing them for the role of gender. We have added the following paragraph to our discussion (beginning on line 454). We thank you for this question as we think it yielded an interesting and important observation. 

“Consideration of adult gender was important across the models. It was then surprising that it was not as big a consideration in relation to the children in the majority of the models. However, where gender was considered, it was important. Illustratively, in their model, Molborn & Lawrence [84] draw attention to the overall weakening of socioeconomic disparities in health lifestyles and a strengthening of gender disparities as children age. Niermann et al [96] model gender differences in the association between family functioning and weight status. While a higher level of family functioning was associated with decreased likelihood of being overweight among girls, this was not the case for boys. In the 2018 model by Shapiro et al., [113] there was a significant association between child's gender and Precaution, Adoption, Process Model (PAPM) stage of decision-making, with parents of boys more likely to report being in earlier PAPM stages. Here, parents of daughters (compared to sons), parents of older children, and parents with a health care provider recommendation had decreased odds of being in any earlier PAPM stage as compared to the last PAPM stage (i.e. vaccinated). None of the models made any room for gender diversity or non-binary gender. We would expect as models of the family continue to evolve, attention to non-binary gender among all family members will become much more prominent in future models.”

Overall, we feel that both of these reviewers have helped us to improve this manuscript and we are very grateful.

---

## [Editor Report · Decision Letter 1]

24 Mar 2021

Family as a health promotion setting: 

A scoping review of conceptual models of the health-promoting family

PONE-D-20-23373R1

Dear Dr. Michaelson,

We’re pleased to inform you that your manuscript has been judged scientifically suitable for publication and will be formally accepted for publication once it meets all outstanding technical requirements.

Kind regards,

Johnson Chun-Sing Cheung, D.S.W.

Academic Editor

PLOS ONE

---

## [Editor Report · Acceptance letter]

31 Mar 2021

PONE-D-20-23373R1 

Family as a health promotion setting:A scoping review of conceptual models of the health-promoting family 

Dear Dr. Michaelson:

I'm pleased to inform you that your manuscript has been deemed suitable for publication in PLOS ONE. Congratulations! Your manuscript is now with our production department. 

Kind regards, 

on behalf of

Dr. Johnson Chun-Sing Cheung 

Academic Editor

PLOS ONE